# Tunable metasurfaces via the humidity responsive swelling of single-step imprinted polyvinyl alcohol nanostructures

Byoungsu Ko[1,5], Trevon Badloe [1,5], Younghwan Yang [1,5], Jeonghoon Park [1], Jaekyung Kim[1], Heonyeong Jeong[1], Chunghwan Jung[2] & Junsuk Rho [1,2,3,4] ✉

The application of hydrogels in nanophotonics has been restricted due to their low fabrication feasibility and refractive index. Nevertheless, their elasticity and strength are attractive properties for use in flexible, wearable-devices, and their swelling characteristics in response to the relative humidity highlight their potential for use in tunable nanophotonics. We investigate the use of nanostructured polyvinyl alcohol (PVA) using a one-step nanoimprinting technique for tunable and erasable optical security metasurfaces with multiplexed structural coloration and metaholography. The resolution of the PVA nanoimprinting reaches sub-100 nm, with aspect ratios approaching 10. In response to changes in the relative humidity, the PVA nanostructures swell by up to ~35.5%, providing precise wavefront manipulation of visible light. Here, we demonstrate various highly-secure multiplexed optical encryption metasurfaces to display, hide, or destroy encrypted information based on the relative humidity both irreversibly and reversibly.

As applications and demonstrations of photonic systems are continuously advancing and improving, the search for interesting functional materials is never-ending. From the plasmonic responses of metallic materials[1] to the strong electric field confinement of high-index dielectric materials[2], various concepts and developments continue to be uncovered. Metasurfaces, 2D arrays of subwavelength sized structures, known as meta-atoms, have been proven as an effective method of controlling the properties of light. Recently, there has been a significant drive towards producing metasurfaces with tunable properties through a variety of different mechanisms[3].

Chemical reactions such as the oxidation and hydrogenation of metals provide an interesting route for tunability but are often fairly slow and require special conditions that are difficult to control[4–7]. Meanwhile, phase change materials such as $Ge_2Sb_2Te_5$[8,9], $VO_2$[10–13], and $Sb_2S_3$[14] have taken the forefront as candidates for tunable photonic materials. Liquid crystals have also been introduced to actively manipulate the optical responses of metasurfaces[15–19]. Each platform

has its own respective benefits and drawbacks, with no obvious stand out solution.

Another proposed class of materials for tunable photonics are hydrogels, polymers that swell under the influence of water ($H_2O$), that have been extensively utilized in medical applications due to their low toxicity and biocompatibility[20,21]. Notable examples include silk[22–24], chitosan[25–27], and polyvinyl alcohol (PVA). They possess extremely interesting characteristics for use in active photonic applications, such as being able to form flexible and strong films. However, the optical properties of PVA have somewhat limited it to use as humidity sensor by changing optical path length[28], due to the fairly low refractive index, especially for applications at visible frequencies. Recently, imaging multiplexing has been realized with PVA-based nanocavities, however, the size of each superpixel in the metasurface is much larger than the wavelengths of visible light, resulting in low holographic quality[29]. A few nanofabrication methods with hydrogels have been investigated, such as for the use as electron-beam lithography (EBL) resist[30], and

[1]Department of Mechanical Engineering, Pohang University of Science and Technology (POSTECH), Pohang 37673, Republic of Korea. [2]Department of Chemical Engineering, Pohang University of Science and Technology (POSTECH), Pohang 37673, Republic of Korea. [3]POSCO-POSTECH-RIST Convergence Research Center for Flat Optics and Metaphotonics, Pohang 37673, Republic of Korea. [4]National Institute of Nanomaterials Technology (NINT), Pohang 37673, Republic of Korea. [5]These authors contributed equally: Byoungsu Ko, Trevon Badloe, Younghwan Yang. ✉e-mail: jsrho@postech.ac.kr

with the addition of a platinum (Pt) coating on an aluminum reflector, a metal-PVA-metal multilayer structure has been proven for tunable structural color using grayscale EBL[31]. However, the usage of PVA for photonic devices has been limited since hydrogels typically have low thermal-resistance, an essential property for post-processing during fabrication. Using hydrogels such as PVA for optical metasurfaces provides another option for tunable, flat optical devices. However, to realize all of the benefits that metasurfaces offer, such as multiplexing various functionalities through phase and amplitude manipulation of the incident wavefront, the meta-atoms must first be tall enough to accumulate enough phase, and second must be nanoscale in size in order to operate at visible wavelengths.

Here, we report subwavelength, high-aspect ratio structured PVA for large-scale, flexible, and tunable metasurfaces. We characterize the mechanical and optical properties of PVA thin films and uncover the amount of swelling when exposed to different relative humidity (RH) conditions. We then prove the ability of patterning large-scale, high-resolution PVA meta-atoms for metasurfaces using one-step nanoimprint lithography (NIL). We finally exploit the swelling characteristics of the PVA meta-atoms to develop three distinct metasurfaces by multiplexing reflected nanoprints under white light illumination and holographic images in transmission when illuminated with coherent light. First, we demonstrate irreversible PVA metasurfaces with potential applications in the food industry and biomedical fields as non-toxic single-use smart labels, which could be applied to any surface such as curved plastic or glass bottles. Such an intelligent packaging for food products could help to improve food safety[32,33], as well as add a measure of safety for authentication against counterfeits[34], while smart sensors in the biomedical field could have a profound impact for biosensors and bioimagings[32,33]. Second, we demonstrate an irreversible optical security system that alerts the receiver if their sensitive data has been compromised by destroying the holographic and color print information after the hidden color print is uncovered. Finally, by depositing an ultrathin (10 nm) platinum (Pt) layer on the nanoimprinted PVA, we demonstrate a platform for continuously reversible PVA metasurfaces that could be used repeatedly for humidity sensitive optical security. We envision that

metasurfaces based on the tunable properties of hydrogels that have physical and optical characteristics that are manipulated by RH could be applied in a variety of fields, such as optical security and sensing, as well as being impactful in biochemical applications.

## Results
### Characterization of PVA films

To explore the possibilities of using PVA as a material for tunable nanophotonics, we first investigate the swelling and optical properties of spin-coated PVA thin films. PVA consists of a porous three-dimensional network structure with crosslinking between polymer chains (Fig. 1a)[30,35]. We prepare aqueous PVA solutions by dissolving solid PVA powder into deionized water with concentrations of 3 and 5 wt% (Fig. 1b). These concentrations can be simply spin-coated to form thin films with subwavelength thicknesses. The thickness of the PVA films is inversely proportional to the root of the rotation speed, and also increases as the concentration increases. The minimum thickness approaches 67 nm at 4000 rpm with a PVA solution of 3 wt%, while films with a thickness of 454 nm are produced at 2000 rpm with 5 wt% (Supplementary Note 1)[36]. The presence of side hydrophilic hydroxyl (OH) groups allows the PVA to swell and dissolve in $H_2O$. At high RH, more $H_2O$ molecules are dispersed in the surrounding air, promoting more hydrogen bonds in the side OH group of the PVA. Therefore, the thickness of the PVA thin films changes in response to the RH of the surrounding air (Fig. 1c). This absorption results in an increase of the physical size of the hydrogel thin film. Therefore, the higher the PVA concentration in the thin film, the more PVA molecules that can absorb $H_2O$ exist, resulting in an increase in the maximum amount of swelling. It is crucial to quantify the tunable properties of PVA to employ it in nanophotonic applications, so the change of thickness thin films made with 3 wt% PVA is measured under different RH conditions using an atomic force microscope (AFM) equipped with a wireless humidifier (Fig. 1d). The measured swelling for the film spin-coated at 1000 rpm reaches ~35.5% increase with a dramatic increase in thickness from RH 25 to 80%. Meanwhile the film spin-coated at 3000 rpm shows a smaller total swelling of only 27.5%, with a more uniform increase with respect to RH. The increase in thickness is

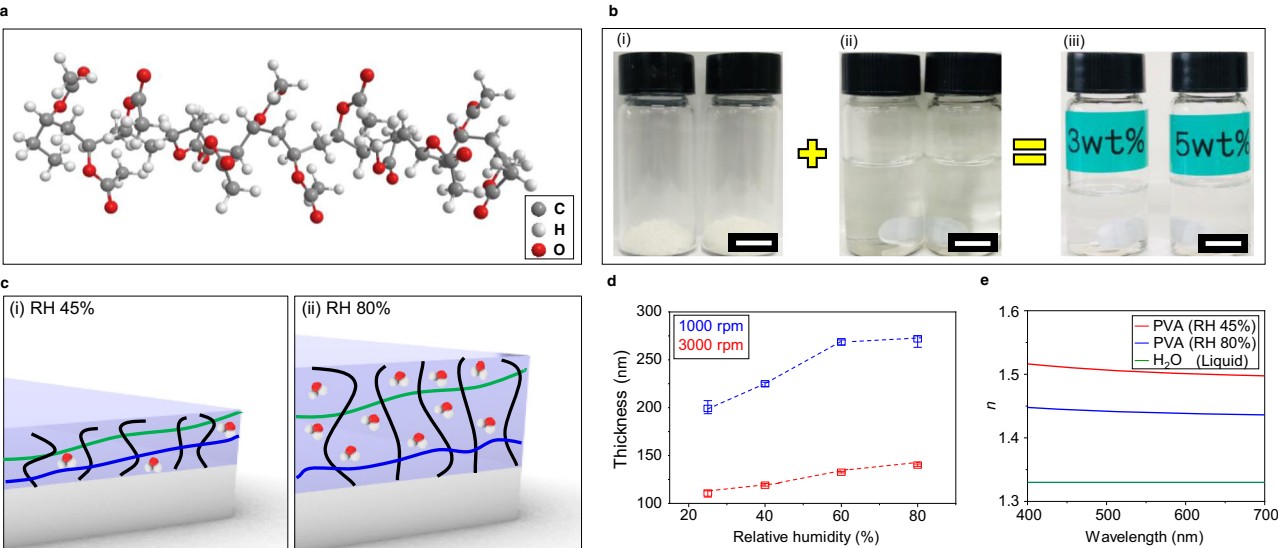

**Fig. 1 | Characterization of the swelling and optical properties of polyvinyl alcohol thin films. a** Schematic of the molecular structure of polyvinyl alcohol (PVA). **b** Aqueous PVA solutions are created by dissolving (i) solid PVA powder into (ii) deionized water with concentrations of (iii) 3 and 5 wt%. All scale bars: 5 mm. **c** Schematic illustration of the swelling mechanism of PVA. As the relative humidity (RH) increases from (i) RH 45 to (ii) 80% the amount of $H_2O$ molecules absorbed

into the hydrogel increase, leading to an increase in thin film thickness. **d** Measured thickness of PVA thin films spin-coated at 1000 (blue) and 3000 rpm (red) at various RH. **e** Measured and calculated refractive index of PVA at RH 45 and 80%, respectively. The refractive index of PVA at RH 80% is calculated using effective medium theory.

generally uniform across the whole film (Supplementary Note 2). As is typical for polymers, the bandgaps of PVA are located in the ultraviolet and infrared regions, while exhibiting zero-optical losses over the full-visible spectrum. This means that PVA is a visibly transparent material before and after spin coating into thin films. We confirm the latter experimentally, by measuring the transmission of PVA thin-films as shown in Supplementary Note 3. The refractive index ($n$) of PVA is measured in the visible region (from 400 to 700 nm) using ellipsometry at a RH of 45% (Fig. 1e and Supplementary Note 4). Furthermore, using effective medium theory, we calculate the $n$ of PVA at RH 80% (Supplementary Note 5). Since the $n$ of water is lower than that of PVA, the $n$ of the swollen PVA at RH 80% slightly decreases.

## One-step NIL of PVA metasurfaces

Optical metasurfaces that operate at visible wavelengths require meta-atoms with subwavelength dimensions to interact with and manipulate the incident light[37,38]. By employing PVA as the nanoscale meta-atoms, we open up a tunable degree of freedom through the swelling and deformation of the meta-atoms with relation to RH various nanofabrication techniques are generally used to create metasurfaces, such as EBL, focused ion beam lithography, and direct laser writing[39]. However, these methods suffer from the inherent drawbacks of low-throughput and difficulties when fabricating nanostructures over large wafers due to the numerous processes required to finalize the sample. Recently, breakthroughs in NIL have demonstrated a method of realizing large scale metasurfaces without the need for secondary processes such as etching[40–42]. The PVA film thickness can be controlled by the rotation speed during spin-coating and by changing the concentration of the PVA. Using a low angular-velocity during spin-coating produces non-uniform PVA films, which are also unsuitable for NIL (Supplementary Note 6), while solutions with a higher concentration of PVA have a higher viscosity and tend to stick to the mold during the NIL process, so are therefore not recommended for NIL (Supplementary Note 7). Here, we use PVA thin-films that are deposited using an angular velocity of 3000 rpm with 3 wt% PVA to fabricate functional metasurfaces using a one-step NIL process.

The imprinting process starts with the replication of the master mold. Polydimethylsiloxane (PDMS) is commonly used as the intermediate soft stamp for the replication of nanostructure arrays. However, due to its high viscosity, pattern sizes below 100 nm have difficulty penetrating the mold, thereby proving a challenge for precise nanoscale replication[41,43]. Therefore, low viscosity hard-PDMS (h-PDMS) is instead directly coated onto the master mold. Subsequently, to facilitate handling, a PDMS layer is poured and cured on top of the h-PDMS to produce the soft mold. The PVA solution is then coated onto the hydrophobic-processed soft mold and is then transferred to the substrate through mechanical pressure (Fig. 2a). Through the printing process, the nanostructures are formed along with a residual layer of PVA. We use this one-step NIL technique to prove the ability of large area nanofabrication using PVA to create 1D gratings, and arrays of 2D nanostructures and nanoholes (Fig. 2b) with resolution down to the sub-100-nm scale with aspect ratios reaching up to 10 (Supplementary Note 8). This large aspect ratio opens up the potential to create meta-atoms that are tall enough to accumulate enough phase at the subwavelength scale, enabling the potential for local phase encoding that can be exploited for geometric phase metaholography. The different types of structure are fabricated with high fidelity and consistency over the whole substrate, while the master mold can be used repeatedly after cleaning it with pure water. Furthermore, since the NIL technique does not depend on any secondary operations that require any specific conditions such as high temperatures that can limit the choice of substrate, we are able to fabricate PVA metasurfaces directly onto flexible polymer substrates, allowing for the application of PVA-based metasurfaces onto any kind of surface (Fig. 2c). As a demonstration of a potential practical application, we attach PVA

metasurfaces to the curved surface of a glass bottle (Fig. 2d). This substrate selectivity enables various optical applications of PVA-based metasurfaces, and alongside the low toxicity of PVA[44], the potential for use as smart labels on editable products, safe wearable optical security devices, and applications in the biomedical field is profound. Furthermore, we experimentally confirm that the potentially harmful coupling agents used in the fabrication of the soft mold do not exist in the final PVA metasurfaces to prove their potential for such implementations (Supplementary Note 9).

## Erasable PVA-based metasurfaces for smart labels

To design PVA metasurfaces for smart labels, we encode two different pieces of information, one in the public domain that is simply visible through a reflected nanoprint under the illumination of white light, and the other as a hidden holographic image that is displayed under the illumination of a coherent laser source. To achieve this, we directly print PVA metasurfaces on top of a fused silica ($SiO_2$) substrate (Fig. 3a). After the meta-atoms that make up the metasurface have been successfully printed using NIL, the designed optical properties are clearly observed, however, after exposure to high RH conditions (~RH 80%), the nanostructures are completely and irreversibly destroyed (Fig. 3b).

Although the $n$ of PVA is too low to strongly confine the incident light in the nanostructures to produce the Mie resonances that have been generally utilized for structural color metasurfaces[45,46], it is still large enough to manipulate the reflectance spectra depending on the geometry and size (Supplementary Note 10). Meanwhile, to produce holographic images we employ the concept of Pancharatnam-Berry (PB) phase, which is also known as geometric phase[47,48]. This requires the design of anisotropic meta-atoms that act as half-wave plates to convert incident circularly polarized light into its opposite helicity, i.e., converts incident left circularly polarized (LCP) light into right circularly polarized (RCP) light (Supplementary Note 11). Since the reflected color depends on the filling ratio of the meta-atoms, designing two meta-atoms with different geometries that provide the same cross-polarization conversion efficiency ($\eta_{cross}$) allows us to multiplex both a nanoprint and hologram into the same metasurface. To achieve this, we first simulate the $\eta_{cross}$ of rectangular PVA meta-atoms with a periodicity ($P$) of 400 nm, height ($H$) of 500 nm, and lengths ($L$) and widths ($W$) from 20 to 370 nm, at the desired wavelengths of 635, 532, and 450 nm, for red, green, and blue holographic images, respectively. Meta-atoms with the same $L = 320$ nm and different $W = 200$ and 90 nm are chosen and denoted as Meta A and Meta B, respectively. Meta A demonstrates $\eta_{cross}$ of 1.02, 1.88, and 2.01%, while Meta B shows $\eta_{cross}$ of 1.04, 2.03, and 4.45% at the working wavelengths of 635, 532, and 450 nm, respectively (Fig. 3c). We retrieve the required phase profile for the holographic images using the well-known Gerchberg–Saxton (GS) algorithm and implement Meta A and Meta B with the correct rotations at each spatial location to produce the desired nanoprint of a quick response (QR) code when illuminated with unpolarized white light (Fig. 3d), and the holographic image of a key under coherent LCP laser illumination at the designed wavelengths (Fig. 3e). After the metasurfaces are exposed to RH 80%, the meta-atoms expand and aggregate, resulting in the destruction of the nanostructures. A real-time video of the QR code being destroyed is provided (Supplementary Video 1). The information that was stored in both the nanoprint and metahologram is irreversibly destroyed as the meta-atoms lose their nanoscale geometries and form an effective film of PVA. To determine how the physical dimensions of the meta-atoms are modulated through the absorption of $H_2O$ molecules at different RH, we simulate the swelling of Meta A (Supplementary Note 12). Although the color prints appear to show an unevenness for the same meta-atoms, this is due to the use of optical microscopy to image the color prints (Supplementary Note 13).

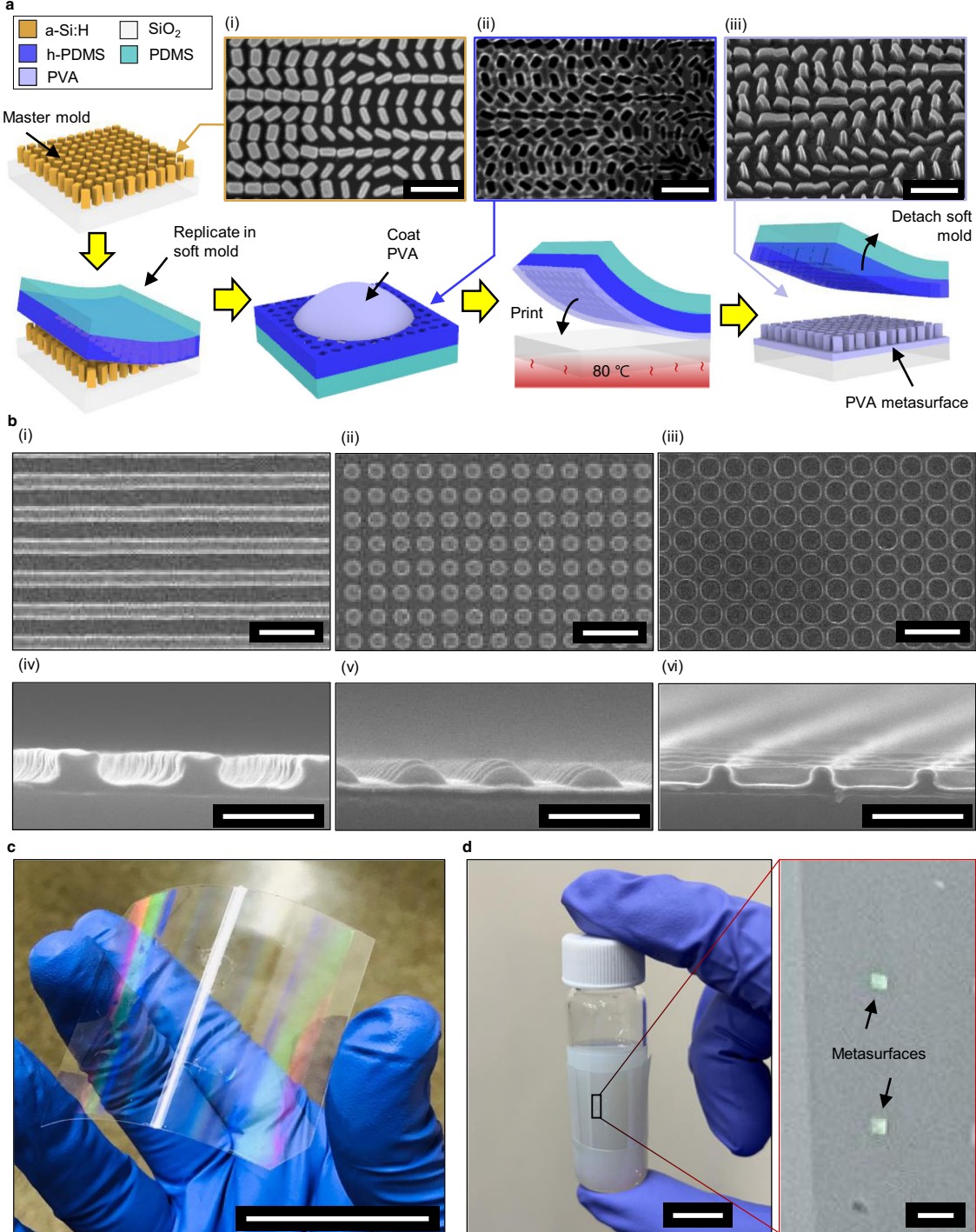

**Fig. 2 | Fabrication of polyvinyl alcohol nanostructures using nanoimprint lithography. a** The nanoimprint lithography (NIL) process for polyvinyl alcohol (PVA). (i) The master mold is fabricated using electron-beam lithography (EBL) and (ii) a negative soft mold is casted using polydimethylsiloxane (PDMS). The soft mold is then coated with PVA, and the NIL process is conducted to produce (iii) PVA metasurfaces. All scale bars: 1 μm (**b**) Scanning electron microscope (SEM) images of (i) 1D gratings, (ii) 2D arrays of nanostructures and (iii) nanoholes fabricated using PVA NIL. Scale bars: (i), (ii), (iii) 1 μm, and (iv), (v), (vi) 500 nm. **c** Photograph of cm-scale PVA nanostructure printed on a flexible polycarbonate (PC) film using NIL. Scale bar: 5 cm. **d** Photographs of the PVA metasurface attached to the curved surface of a glass bottle. Scale bar: 1 cm. Inset: close-up of the 500 × 500 μm² PVA metasurfaces. Inset scale bar: 1 mm.

Such non-toxic and erasable PVA-based metasurfaces could be potentially used for applications such as in smart labels for food packaging, or in the biomedical field. Moreover, the wide substrate selectivity allowing for the fabrication of flexible metasurfaces could be extremely beneficial for such applications, as the packaging is generally not flat. We envision that these PVA metasurface-based smart labels could hide sensitive information in the metahologram, while public information is clearly displayed in the nanoprint. After undesired exposure to RH 80% humidity conditions, both pieces of information are lost, informing the user that the product has been compromised. The high humidity conditions can also be purposefully produced using the breath of a human[49] (Supplementary Note 14), allowing for sensitive information to be destroyed at will after it has been retrieved by the receiver.

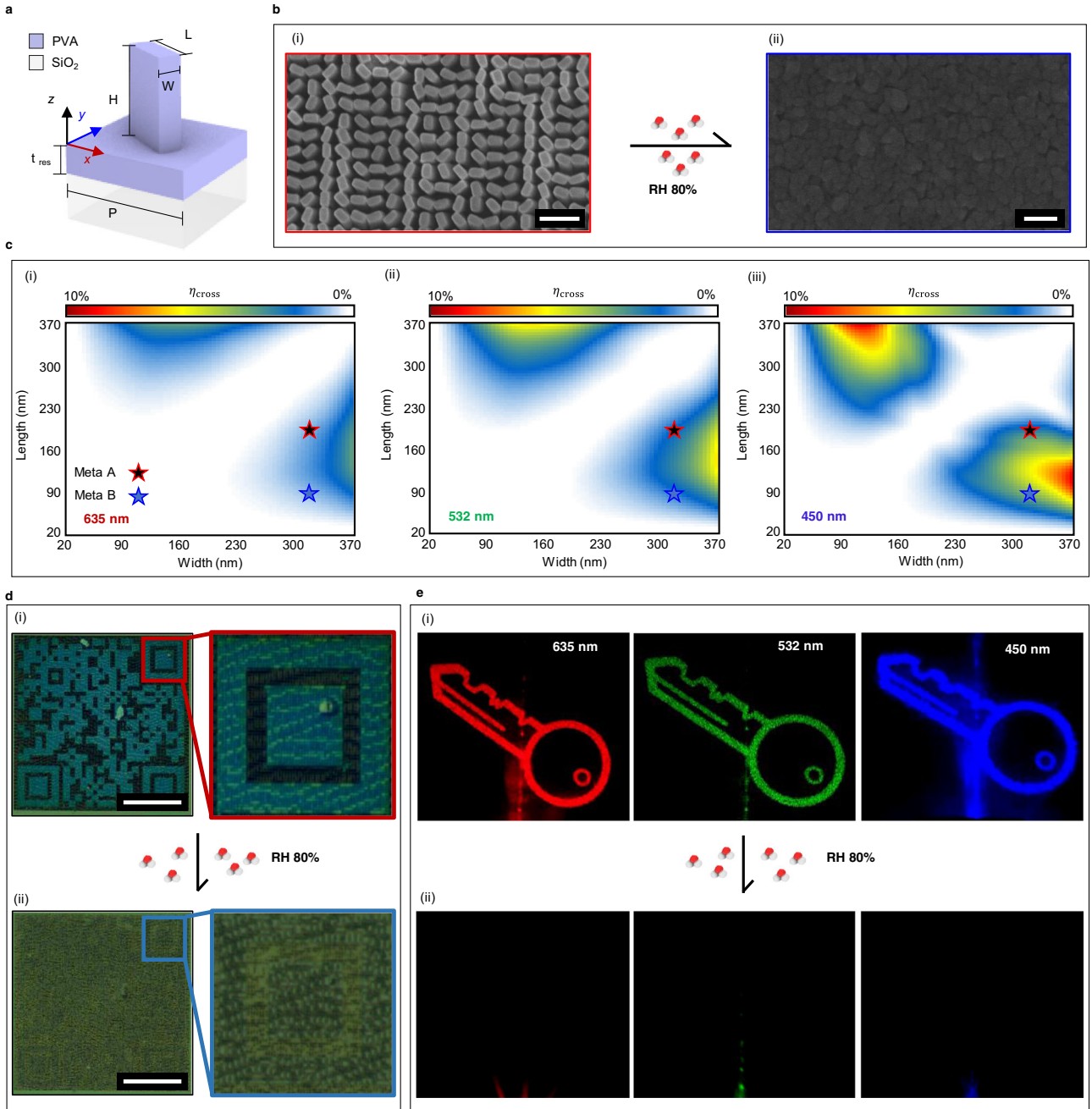

**Fig. 3 | Polyvinyl alcohol metasurfaces for multiplexed structural color and metaholography for humidity sensitive smart labels. a** Schematic illustration of the polyvinyl alcohol (PVA) meta-atom for humidity sensitive smart labels. **b** Scanning electron microscope (SEM) images of the PVA metasurface (i) before and (ii) after exposure to relative humidity (RH) 80% conditions. Scale bars: (i) 1 μm, (ii) 100 nm. **c** Calculated cross-polarization efficiencies at the working wavelengths of (i) 635, (ii) 532, and (iii) 450 nm. Two meta-atoms (denoted as Meta A and Meta B) are selected. Periodicity (*P*) and height (*H*) are fixed at 400 and 500 nm. **d** Reflected optical microscope image of the metasurface under white light (i) before and (ii) after exposure to RH 80% conditions. All scale bars: 100 μm. **e** Photographs of the holographic images produced from the PVA metasurface when illuminated with red, green, and blue lasers with wavelengths of 635, 532, and 450 nm, respectively, (i) before and (ii) after exposure to RH 80% conditions. Both the nanoprint and holographic information are completely destroyed and unrecoverable after exposure to RH 80%.

## Irreversible two-channel humidity sensitive optical security

By printing the PVA metasurface onto a hydrogenated amorphous silicon (a-Si:H) substrate (Fig. 4a), the reflection characteristics are different owing to the considerable extinction coefficient at visible wavelengths (Supplementary Note 15)[50]. We use this to design a highly secure two-channel encryption system by exploiting the fact that the geometry of the meta-atoms is destroyed after exposure to RH 80% conditions. Since the spatial modulation of the phase of the incident light is provided through the meta-atoms geometry to

recreate the holographic images, swelling in response to an increase in RH instantly destroys the holographic information encoded into the metasurface. The reflected color, however, can be designed to reveal hidden information at a specific RH, as Meta A and Meta B swell by different amounts due to their volume difference. At RH 45%, a single uniform color is displayed in reflection, hiding the encrypted information, until it is irreversibly revealed through exposure to RH 80% conditions. After the encoded nanoprint is decrypted, the meta-atoms are destroyed, leaving an

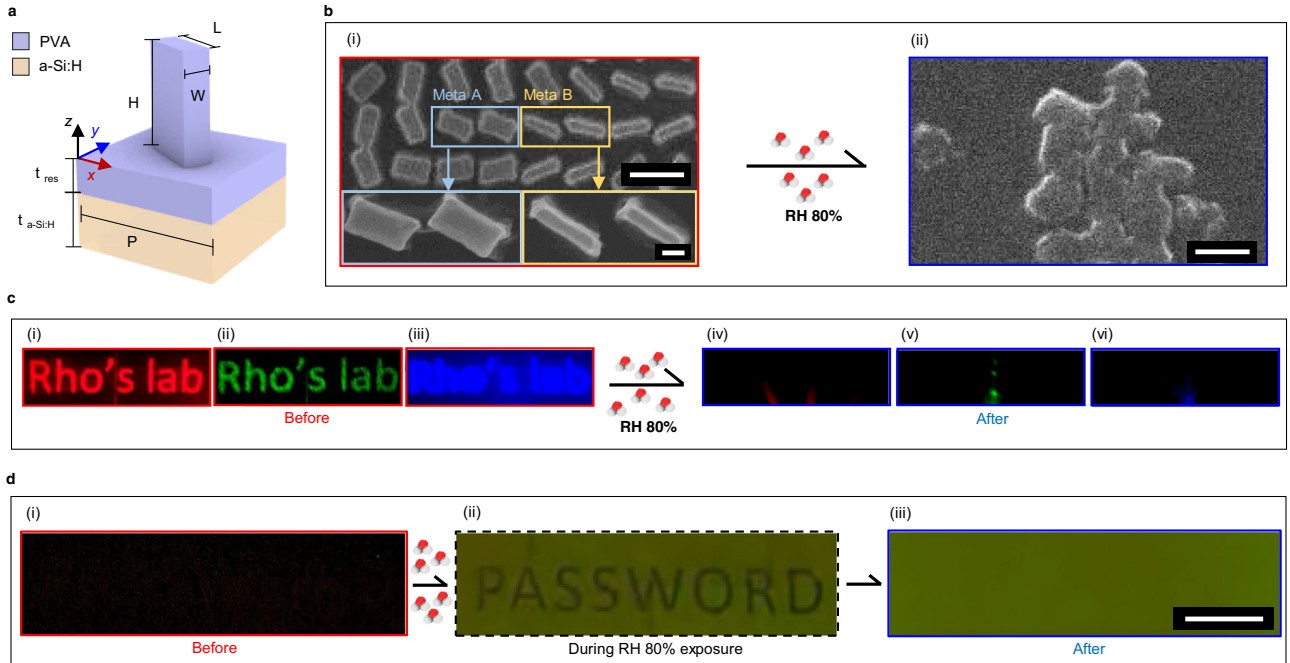

**Fig. 4 | Irreversible polyvinyl alcohol metasurfaces for a single-use two-channel security platform. a** Schematic illustration of a polyvinyl alcohol (PVA) meta-atom on top of hydrogenated amorphous silicon (a-Si:H). **b** Scanning electron microscope (SEM) images of the PVA metasurface (i) before and (ii) after exposure to relative humidity (RH) 80% conditions. Meta A and Meta B are reproduced with high fidelity using nanoimprint lithography (NIL). Scale bars: (i) Top: 500 nm, Bottom: 100 nm, and (ii) 500 nm. **c** Photographs of the holographic images produced from the PVA metasurface when illuminated with red, green, and blue lasers with wavelengths of 635, 532, and 450 nm, respectively, (i), (ii), (iii) before and (iv), (v), (vi) after exposure to RH 80% conditions. The hologram is instantaneously destroyed. **d** Reflected microscope images of the PVA metasurface (i) before, (ii) during, and (iii) after exposure to RH 80% conditions. The encrypted print is uncovered as the meta-atoms swell at different rates. Scale bar: 50 μm.

unstructured thin film of PVA. The receiver of the metasurface can confirm if the metasurface has been compromised by illumination of coherent laser light. If the hidden information has been unwantedly uncovered and stolen, the holographic information will be missing.

The fidelity of the reproduction of Meta A and Meta B through NIL is confirmed through scanning electron microscope (SEM) images (Fig. 4b) where the two distinct meta-atoms are clearly visible. In this undisturbed state, the metasurface produces the encoded holographic images at the three working wavelengths as designed (Fig. 4c). Through exposure to RH 80% conditions using a human breath, since the swelling rate of Meta A and Meta B differs due to their overall volumes, the locations of the two different meta-atoms become clearly visible to uncover the hidden reflected image (Fig. 4d). Here, we encode the word 'PASSWORD' as a proof-of-concept example. After returning back to the original room humidity, the meta-atoms have aggregated, so the holographic information has been destroyed and the encrypted information in the nanoprint is hidden again. A real-time video of both the reflected nanoprint and hologram is provided (Supplementary Video 2). As can be seen in the video, the hidden image is vaguely visible after the first two breaths, and completely destroyed after that, while the holographic information instantly disappears after exposure to RH 80% conditions.

This proposed irreversible two-channel encryption method could have a significant impact on optically variable devices and secure optical information sharing applications. Since revealing the password encrypted in the nanoprint of the humidity sensitive metasurface destroys the encoded holographic information, the lack of a reconstruction of the holographic image instantly notifies the user that their sensitive information may have been compromised, while the information in the color print can only be observed a few times and destroyed at will.

## Reversible two-channel humidity sensitive optical security

Although the irreversible metasurfaces presented up to now can be used for optical security, there are also many situations where it is beneficial for the metasurfaces to be completely reversible and reusable. We demonstrate this reversibility using PVA metasurfaces by depositing a non-oxidizing 10 nm layer of Pt onto the PVA metasurface (Fig. 5a)[51]. This allows the individual meta-atoms to retain their nanoscale structures after the swelling and deswelling process, so both the encrypted print and holographic information can be retained. The optical properties of a Pt metasurface made up of Meta A structures are measured to confirm the existence of any additional effects from the Pt layer (Supplementary Note 16).

We experimentally demonstrate an implementation of this multiplexed 2-channel reversible encryption platform. After the NIL and Pt deposition processes, a uniform color is displayed under the reflection of visible light at RH 45%, while producing a holographic image under the illumination of a coherent laser (Fig. 5b). When exposed to RH 80% using a breath, the hidden word 'PASSWORD' is decrypted temporarily due to the difference in swelling of Meta A and Meta B, while the holographic information is temporarily hidden due to the increased noise in the holographic image (Supplementary Note 17). Thanks to the Pt layer, on the contrary to the previous examples, the meta-atoms preserve their nanoscale geometry after exposure to RH 80%. This allows for the holographic image to be displayed again, and for the color print to be encrypted when the RH is reduced back to room level (Fig. 5c). The decryption is experimentally measured to occur when the RH is around 68% (Supplementary Note 18). The robustness of this reversibility is confirmed by exposing the PVA metasurface to exhaled breath up to 100 times (Fig. 5d). The SEM images show that the meta-atoms morphologies are maintained, which allows for the repeatable decryption of the hidden color image and the recreation of the multiplexed metahologram. Videos of these two reversible processes are provided in Supplementary Video 3 and Supplementary Video 4. This

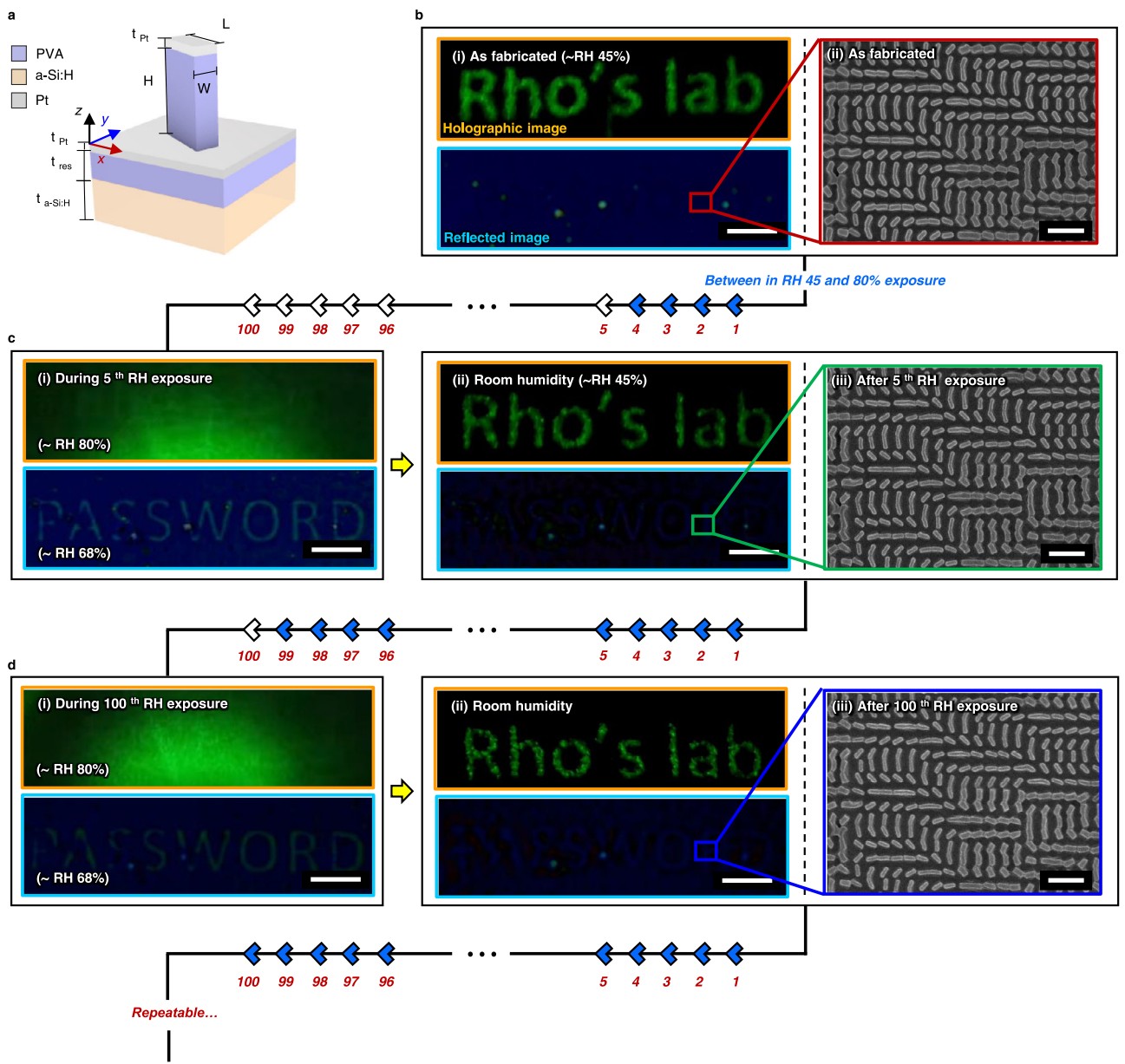

**Fig. 5 | Reversible polyvinyl alcohol metasurfaces for a two-channel security platform. a** Schematic illustration of a platinum (Pt) coated polyvinyl alcohol (PVA) meta-atom on top of hydrogenated amorphous silicon (a-Si:H). **b** (i) Photographs of the holographic, and reflected images of the Pt-coated PVA metasurfaces. Scale bar: 50 μm. (ii) scanning electron microscope (SEM) image of the PVA metasurface, as fabricated. Scale bar: 1 μm. Experimental results after **c** 5 and **d** 100 cycles of exposure to RH 80% using human breath. (i) During and (ii) after RH 80% exposure. All scale bars: 50 μm. (iii) SEM images of the PVA metasurface. All scale bars: 1 μm.

robust, reversible, and repeatable response allows the hidden information in the reflected visible color print to be decrypted at will, which has potential uses for secure information storage. Additionally, we prove the proposed PVA metasurfaces are robust even after exposure to thermal stress of up to 70 °C, the process is maintained for 100 exposures to 80% RH (Supplementary Note 19).

## Discussion

We have investigated the optical properties of PVA as a material for use as meta-atoms in optical metasurfaces that operate over the full visible regime. Furthermore, we utilized a one-step NIL technique to demonstrate all types of high-aspect ratio meta-atom structures using PVA, including nanogratings, nanopillars, and nanoholes, as well as demonstrating the freedom in substrate selection, including SiO₂, a-Si:H, and polymer, which could be useful for various applications where flexible, transparent, or strong substrates are required. Moreover, we experimentally realized tunable metasurfaces using nanoimprinted meta-atoms made of PVA for multiplexed humidity-dependent structural color and holography. We developed three different potential uses of such PVA metasurfaces, namely as single-use humidity sensitive smart labels and irreversible optical security, and by adding a thin layer of Pt, we demonstrated a method of creating a continuously reversible optical security device that is robust for at least 100 cycles, with almost no visible defects at the nanoscale as confirmed by SEM images. By choosing the appropriate substrate and coating, the reflective properties can be modulated, and the nanoscale geometry of the one-step imprinted meta-atoms can be selectively preserved or destroyed through exposure to high humidity conditions, providing an alternative route for actively tunable metasurfaces that can manipulate the

phase and amplitude of incident light at the subwavelength scale. The PVA-based metasurfaces presented here could have an impact in various fields such as optical security and smart labels for food and biological samples due to their ease of fabrication, substrate selectivity, and low toxicity. Rather than requiring humid conditions to erase and destroy the information secured in the metasurface, the user can also simply breathe onto the sample to provide a high enough humidity to achieve the same effect. Although the structural color of the metasurfaces in this work require optical microscopy systems to image, this is not a fundamental limitation of the design, which can be simply scaled-up based on the size of the master mold. By implementing fabrication methods to create centimeter-scale metasurfaces for the master mold, the encrypted color images in the PVA metasurfaces proposed here could easily be read by eye, or using everyday smartphone readers. Furthermore, the one-step nanoimprinting method leads well to potentially high-throughput techniques such as roll-to-roll imprinting. Other hydrogels with humidity-dependent properties such as chitosan could be substituted for the PVA using the methods described here. However, physical properties of each material such as viscosity would require further optimization and development of the nanoimprinting technology to produce the correct conditions and stamps that can be used for large-scale fabrication. The proposed PVA metasurfaces could be vulnerable to harsh weather conditions such as frequent rain and locations with generally high RH. This could be supplemented with protective materials to block direct contact with water, with a trade-off in terms of response time. For locations with constant high humidity, the metasurfaces could be designed with the baseline RH of the specific place in mind and adapted accordingly, while other hydrogels could also be explored.

## Methods

### Numerical simulations

Numerical simulations were performed using the commercially available finite-difference time-domain (FDTD) solver (Lumerical, Ansys). The measured refractive index of PVA was used throughout. All color conversions and calculations were performed using the open-source package 'Colour', in Python, using the CIE 1931 2˚ standard observer and the D50 illuminant. Conversion efficiencies of the meta-atoms were calculated using periodic boundary conditions in the $x$- and $y$-directions, with perfectly matched layers in the $z$-direction. The finite element method solver, COMSOL Multiphysics version 5.6 was used for the swelling simulations.

### Fabrication of master mold

The designed master mold is fabricated using conventional plasma-enhanced chemical vapor deposition, inductively coupled plasma reactive-ion etching and electron-beam lithography (EBL). Firstly, hydrogenated amorphous silicon (a-Si:H) was deposited on a cleaned fused silica (SiO$_2$) substrate with plasma-enhanced chemical vapor deposition. A positive electron-beam resist (Microchem, PMMA 495 A6) was spin-coated at 4000 rpm for 1 min on the a-Si:H. Then, the substrate was baked at 180 °C for 5 min. A conductive polymer (Showa Denko, Espacer 300Z) was spin-coated at 2000 rpm for 1 min to prevent charge accumulation during the EBL process. The coated photoresist was exposed using EBL (Elionix, ELS-7800) at an acceleration voltage of 100 kV. The coated conductive polymer was cleaned with deionized water at room temperature for 2 min, and the exposed resist was developed by immersing it in solution (Microchem, MIBK:IPA = 1:3) at 0 °C for 10 min. A chromium (Cr) mask was deposited using electron beam deposition (KVT, ENS-4004) before lift-off of the undeveloped photoresist layer. The Cr masks were transferred to the a-Si:H after inductively coupled plasma reactive-ion etching of the a-Si:H and Cr-mask etching with CR-7.

### Preparation of PVA solution

The hydrolyzed commercial PVA (Dongyang Syn Co.) molecular weight of 17,000 granular particles was dissolved in deionized water at 90 °C.

### Replication process of the master mold

The master mold was treated with a vaporized silane coupling agent (Sigma Aldrich, Trichloro-1H, 1H, 2H, 2H-Perfluorooctyl-silane) to form a surface self-assembled monolayer (SAM) for 10 min at 140 °C to improve the demolding. The h-PDMS solution was prepared by mixing 1.7 g of vinylmethyl copolymers (Gelest, VDT-731), 9 µL of platinum-catalyst (Gelest, SIP6831.2), 0.05 g of the modulator (Sigma Aldrich, 2,4,6,8-Tetramethyl-2,4,6,8-tetravinylcyclotetrasiloxane), 1 g of toluene, and 0.5 g of siloxane-based silane reducing agent (Gelest, HMS-301). The h-PDMS was spin-coated on the master mold at 2000 rpm for 60 s, and then baked at 80 °C for 2 h. A degassed mixture 10:1 weight ratio of a PDMS (Dow corning, Sylgard 184 A) and curing agent (Dow corning, Sylgard 184 B) was poured on the h-PDMS layer then baked at 80 °C for 3 h. The soft mold was demolded carefully from the master mold, then treated identically to a self-assembled monolayer (SAM) as reported above.

### Printing of PVA-based metasurface

The 3 wt% of PVA solution was spin-coated onto the soft mold, heated up to 80 °C then imprinted on the substrate for 10 min at 2 bar. The soft mold was completely released from the substrate, leaving the PVA metasurface imprinted on the substrate. The water-soluble PVA allows for the continuous reuse of the soft molds. After the NIL process, the soft mold can be reused by washing away the leftover PVA residue with deionized water and conducting O$_2$ plasma and SAM treatments.

### Pt-coated for reversible PVA metasurface

The top ultrathin (10 nm) platinum (Pt) was deposited by commercial Pt coater (Hitachi, MCI 1000). The deposition process was conducted at 20 mA for 60 s to form a 10 nm ultrathin film onto the PVA metasurface.

### Characterization of the refractive index of PVA

The refractive index of PVA in the visible region (from 400 to 700 nm) was measured using ellipsometry (J.A. Woollam, M-2000D) at clean room humidity conditions (-RH 45%). A deuterium/halogen light source was illuminated onto the sample at an incident angle of 65°, and as standard for visibly transparent polymers, the Cauchy model was used to determine the refractive index of the PVA films. The measured refractive index was used throughout in simulations for the meta-atom design. The effective refractive index at RH 80% was calculated using the refractive index of H$_2$O = 1.33.

### Swelling characterization

To understand the swelling of PVA, the thickness changes in PVA film at a varied humidity (RH 25, 45, 65, 80%) were measured using an atomic force microscope (AFM) (Park systems, XE7) at room temperature (20 ± 2 °C). The RH in the AFM chamber was controlled with a wireless humidifier from dry conditions (RH 25) to 45, 65, and 80%, and real-time humidity changes were observed with a mobile humidity meter (Lutron, HT-3007SD). To achieve RH 25%, which is lower than the atmospheric humidity, silica gel was employed to capture the excess humidity in the chamber. When the humidity reached the desired condition, the thickness was measured.

### Optical measurement

The nanoprint images were captured with an optical microscopy system (Olympus, MX63) composed of a CCD camera (Lumenera, Infinity2-2). The nanoprint images were observed with a ×50/0.80 objective lens. To demonstrate the dynamic tuning process, human

breath was directly exhaled onto the sample at a distance of around 2.5 cm for around 1.5 s per breath. The switching of imaging, which occurs by the swelling/deswelling of the PVA metasurface was captured in real-time during the reversible transition of humidity between low (RH 45%) and high (RH 80%). The holographic images were recorded using a home-made optical setup. The beams were generated using lasers (Thorlabs, diode-pumped solid-state lasers) with wavelengths of 450 (blue), 532 (green), and 635 nm (red), respectively. The beam sequentially passed through a linear polarizer (Thorlabs, Ø 1/2″ Unmounted linear polarizers), and a 45°-rotated quarter-wave plate (Thorlabs, Ø 1/2″ Mounted achromatic quarter-wave plates) to form a left circular polarized light beam. A 300 μm diameter pinhole (Thorlabs, Ø 1″ Mounted Pinhole) was added to block out any unnecessary light. The holographic images were projected onto a screen 5 cm away from the metasurfaces and captured using a smartphone camera (Samsung electronics, Galaxy S9).

## Data availability
The data that support the findings of this study are available from the corresponding author upon request.

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

## Acknowledgements

This work was financially supported by the Samsung Research Funding & Incubation Center for Future Technology grant (SRFC-IT1901-52) funded by Samsung Electronics, the POSCO-POSTECH-RIST Convergence Research Center program funded by POSCO, and the National Research Foundation (NRF) grant (NRF-2022M3C1A3081312) funded by the Ministry of Science and ICT, Republic of Korea. B.K. acknowledge the NRF fellowship (NRF-2022R1A6A3A13066244) funded by the Ministry of Education (MOE), Republic of Korea. Y.Y. acknowledge the Hyundai Motor *Chung Mong-Koo* fellowship, and the NRF fellowship (NRF-2021R1A6A3A13038935) funded by the MOE, Republic of Korea. The authors thank Prof. Young Min Song and Mr. Joo Hwan Ko (Gwangju Institute of Science and Technology) for equipment support for the humid-related experiments.

## Author contributions

J.R. and B.K. conceived the idea and initiated the project. B.K. fabricated the PVA metasurfaces using nanoimprint and measured the mechanical and optical properties. T.B. performed the numerical calculations and developed the codes. Y.Y. designed the metaholograms, calculated optical responses of the meta-atoms, fabricated the master molds, and measured the holographic and QR code images. J.P. simulated and analyzed swelling mechanism of polyvinyl alcohol. H.J. and C.J. assisted in measurements. J.K. assisted in nanoimprinting. All authors contributed to the discussion, analysis, and writing of the manuscript, and gave approval of the final version. J.R. guided the entire project.

## Competing interests

The authors declare no competing interests.
