## [Peer review file · Nature Communications]

REVIEWER COMMENTS

Reviewer #1 (Remarks to the Author):

This paper reports the fabrication of nanostructured polyvinyl alcohol (PVA) films with metasurfaces using nanoimprint lithography. Their structural and optical properties were experimentally and theoretically investigated. Their feasibility for tunable and erasable optical security labels with multiplexed structural coloration and holography functionalities have successfully demonstrated by adjusting relative humidity. The results of this work are interesting and significant to a broad range of readers for Nature Communications, including materials scientists, nano/optical/security engineers, and physicists. However, there are some critical issues that are not fully addressed. To further improve this paper, I have some comments as follows:

1. In Introduction, to show the noteworthy of this work, I kindly recommend that the previous papers on non-toxic single-use (one-time) smart labels or optical security tags and their food industry and biomedical applications, including a brief explanation, should be provided.

2. In section 2.1,

1) Please provide the swelling mechanism of PVA in a condition of high concentration humidity.

2) The authors mentioned that PVA is a visibly transparent material, but there is no data for the transmission of PVA films with and without metasurfaces.

3) Can the authors provide detailed information (light source, incident angles, and calculation models) on how the refractive index (n) of PVA is obtained by using ellipsometry?

4) In the caption of Fig. 1e, the high RH value is 80%, but 85% in the manuscript. Please check this.

3. In section 2.3.1,

1) To numerically examine the effect of water molecules' absorption on the deformation of PVA, the authors performed the thermal expansion simulation of the nanostructured PVA film using commercially available software, COMSOL Multiphysics. Is there a special reason why the swelling property of PVA by humidity is analyzed in terms of thermal expansion?

2) What is the value of the relative humidity when the human breath is applied?

4. It is quite impressive to use the variation in optical phenomena caused by the humidity sensitivity of PVA for security label applications. However, it would be considered to be quite vulnerable to keep the

original status in areas with frequent rains or high humidity. Is there a way to supplement the proposed technology for this?

5. Polymers are relatively weak to heat. How is the thermal stability of the proposed labels?

6. In section 2.3.3.,

1) When a platinum (Pt) thin film is deposited on the surface of PVA films, what is their transmittance? Is there an optical absorption (or plasmonic effect) by the Pt thin film?

2) Continued with the comment #3-2), can the relationship between relative humidity and image variation be quantified in the repeated process? This can provide the threshold of the relative humidity for the decryption of encrypted information.

3) How many times are the reversible and repeatable process of the label kept?

7. Can the authors show the biological safety of the proposed final PVA labels to confirm their low toxicity and biocompatibility, not referring to the previous literature? In the fabrication process, some toxic chemicals including silane coupling agent (Trichloro-1H, 1H, 2H, 2H-Perfluorooctyl-silane) and toluene are used.

8. The sizes of multiplexed structural coloration and holography labels are in the range of a number or tens of a micron. In addition, most images are measured by an optical microscopy system. Such a bulky reading system would be a limitation of the proposed technology in practical applications. Can the labels be further scaled up or be read by easy-accessible readers (e.g., smartphones)?

9. Minor issues:

1) Please double check the scale bars or values of the scale bar in all figures and figure captions. In particular, the scale bars in Fig. 5c should be smaller than 50 μm .

2) Please provide the specific values of the high RH concentrations in the figure captions and manuscript.

3) In Table S1 of Supplementary Note 2, please provide the standard deviation of measured thickness for spin-coated PVA films.

4) In Supplementary Note 7, please check the value of $\sim 96\%$ in the reflectance spectra of a bare SiO_2 substrate. It seems to be $\sim 4\%$.

5) In Supplementary Note 7, what are the evident peaks in the reflectance spectra for the nanocuboid and nanocylinder with width or diameter values larger than 200 nm?

6) Overall, there are mistakes on grammar and spelling in the manuscript. Fine/minor spell check in the English language and style should be required.

Reviewer #2 (Remarks to the Author):

In this manuscript entitled “Tunable metasurfaces via the humidity responsive swelling of single-step imprinted polyvinyl alcohol nanostructures”, the authors have experimentally demonstrated tunable polyvinyl alcohol (PVA) metasurfaces by using the nanoimprinting technique. Owing to the humidity-sensitivity of hydrogel, the PVA nanostructures swell and aggregate under the high relative humidity (RH) conditions, and lead to optical information destruction. Furthermore, by depositing 10 nm Pt onto the PVA metasurface, the top metal preserves the PVA nanostructures from being completely destroyed under the high RH and therefore provides a reversible strategy for two-channel optical information switching.

However, after a careful review of this work, I think that the demonstration of this work is more like a progressive work from their recent publication in Science Advances (Jung, C., Kim, S.J., Jang, J., Ko, J.H., Kim, D., Ko, B., Song, Y.M., Hong, S.H. and Rho, J., 2022. Disordered-nanoparticle-based etalon for ultrafast humidity-responsive colourimetric sensors and anti-counterfeiting displays. Science advances, 8(10), p.eabm8598.). The departure from that work is not big enough and makes this submission lack the novelty and impact to be suitable for publication in Nature communications. The proposed uses of PVA metasurface, such as single-use smart labels and irreversible optical security, are designed by a conventional method and have been demonstrated before similarly. Additionally, in my humble opinion, the PVA metasurface demonstrated in this manuscript is hard to be applied in daily life due to the uncontrollable ambient humidity. When the ambient humidity around the sample is higher than the damage threshold, the encrypted optical information might be automatically destroyed, thus making manual decryption impossible. In addition, the author discussed quite in detail about fabrication processing of PVA films and patterns, therefore I think the paper is rather more appropriate to narrow audiences and is more suitable for specific topics in some material-focused journals.

Here are some further comments in detail for improvement or correction.

1. For the demonstration of large-scale fabrication, the PVA metasurface is printed on a flexible substrate, as shown in Figure 2c. The author should clarify that the structures are fabricated on the whole substrate or part of it.

2. In Figure 3c, the η_{cross} of Meta A and Meta B seem to locate at a similar colour level in the mapping at the wavelength of 635 nm. In fact, the η_{cross} of Meta B (4.45) is twice as the Meta A (2.01). To make the data presentation more scientific and make sense, the numerical value of the colour bar should be provided rather than normalizing it to the maximum and minimum.

3. In Figure 3d(i), the optical microscope image of the QR code shows that the blue area has some green components. Please comment on the colour unevenness.

4. Figure 5 shows the reversible holography by changing the relative humidity (RH). However, the working mechanism seems unclear. Why do the PVA structures swelling cause the holographic image to disappear? The authors should provide more explanation and discussion about the principle of this demonstration.

5. The authors claim that the hidden word is decrypted under the high RH, while the holographic image is hidden. After analyzing Supplementary Video 3, I think this looks misleading. From the video, it could be observed that the nanoprinting image and holographic image can appear/disappear simultaneously during breathing. The authors should carefully address the association between image channel switching and humidity tuning.

6. Regarding the geometry change of PVA nanostructure after being exposed to the high RH, the collapse of most structures (Figure 5c) would drastically influence the conversion efficiency. However, the holography seems to be unaffected. The authors should clarify this and add the discussion of the holographic efficiency.

Reviewer 1

This paper reports the fabrication of nanostructured polyvinyl alcohol (PVA) films with metasurfaces using nanoimprint lithography. Their structural and optical properties were experimentally and theoretically investigated. Their feasibility for tunable and erasable optical security labels with multiplexed structural coloration and holography functionalities have successfully demonstrated by adjusting relative humidity. The results of this work are interesting and significant to a broad range of readers for Nature Communications, including materials scientists, nano/optical/security engineers, and physicists. However, there are some critical issues that are not fully addressed. To further improve this paper, I have some comments as follows:

Response:

We thank the reviewer for their time and effort in carefully reviewing our manuscript. We appreciate that the reviewer deems the results of our work to be interesting and significant to the broad readership of *Nature Communications*. We are extremely grateful for the constructive criticism of our work and for the critical points raised that have allowed us to dramatically increase the quality of our work. We hope that the revisions and additions to our manuscript have fully addressed the issues that the reviewer raised, and that they can recommend our work for publication in *Nature Communications*. We have provided a point-by-point response to each of the concerns of the reviewer and copied the revisions and additions to the relevant replies below, for the convenience of the reviewer.

Response to Reviewer 1**Comment 1:**

In Introduction, to show the noteworthy of this work, I kindly recommend that the previous papers on non-toxic single-use (one-time) smart labels or optical security tags and their food industry and biomedical applications, including a brief explanation, should be provided.

Response 1:

We thank the reviewer for their constructive comment. We agree that adding previous studies about non-toxic smart labels that have been demonstrated for the food industry and biomedical applications would be of great use for general readers, as well as highlighting the usefulness of our work. We have therefore added the following to the introduction, along with some references to research and interesting review articles on the subject, as follows:

In the manuscript:

Line 66: “First, we demonstrate irreversible PVA metasurfaces with potential applications in the food industry and biomedical fields as non-toxic single-use smart labels, which could be applied to any surface such as curved plastic or glass bottles. Such intelligent packaging for food products could help to improve food safety^{32,33}, as well as acting as a measure of safety for authentication against counterfeits³⁴, while smart sensors in the biomedical field could have a profound impact for biosensors and bioimaging^{32,35}.”

Comment 2:

- a) Please provide the swelling mechanism of PVA in a condition of high concentration humidity.
- b) The authors mentioned that PVA is a visibly transparent material, but there is no data for the transmission of PVA films with and without metasurfaces.
- c) Can the authors provide detailed information (light source, incident angles, and calculation models) on how the refractive index (n) of PVA is obtained by using ellipsometry?
- d) In the caption of Fig. 1e, the high RH value is 80%, but 85% in the manuscript. Please check this.

Response 2:

We thank the reviewer for their constructive comments. We have added specific descriptions of the physical properties of PVA films through additional experiments, and we have added the details of the measurement methods to enrich the understanding of the reader. We have also double-checked all the values in the revised manuscript including the RH values. The changes and additions are as follows:

a)

We have added a specific description of the swelling mechanism at the high concentration humidity.

In the manuscript:

Line 92: “The presence of side hydrophilic hydroxyl (OH) groups allows the PVA to swell and dissolve in H₂O. At high RH, more H₂O molecules are dispersed in the surrounding air, promoting more hydrogen bonds in the side OH group of the PVA. Therefore, the thickness of the PVA thin films changes in response to the RH of the surrounding air (Fig. 1c).”

b)

To clarify our statement of PVA films being transparent we conducted further experiments. We fabricated and measured the transmittance of PVA thin-films, as well as a PVA film with

a nanoimprinted metasurface. As discussed in Supplementary Note 4, the uniformity of the PVA thin-films depends on both the concentration of the PVA nanoparticles, and the speed of the spin-coating process. We choose two films to demonstrate the transparent nature of the PVA thin-films, both on glass substrates. The first, spin-coated at 3000 rpm for 60 s with a 3 wt% PVA aqueous solution produces a PVA thin-film of ~ 100 nm. The second, spin-coated at 1000 rpm with a concentration of 5 wt% PVA creates a PVA thin-film of ~ 500 nm. These two thicknesses represent the constituent parts of the metasurfaces in our work, i.e., the thicknesses of the residue layer (~ 100 nm), and meta-atoms (~ 500 nm). Finally, we nanoimprint a metasurface onto a glass substrate under the same conditions as the sample in Figure 3 of the main manuscript. We also measure the transmission of the bare glass substrate as a control sample for comparison. Furthermore, to visually demonstrate the transparency, we take photographs of the two PVA thin-films placed over images of our school logo. These results have been added to the Supplementary Information as Supplementary Note 3, and discussed in the main manuscript as follows:

In the manuscript:

Line 108: “This means that PVA is a visibly transparent material before and after spin coating into thin films. We confirm the latter experimentally, by measuring the transmission of PVA thin-films as shown in Supplementary Note 3.”

In the Supplementary Information:

“Supplementary Note 3: Transparency of PVA thin-films

Figure S3. Transmittance spectra of PVA film & metasurfaces. (a) The measured transmittance comparison of bare glass, thickness variation PVA films, and presence of metasurface array. (b) Photograph of non-coated bare glass (black), the PVA films spin-coated at 3,000 rpm with 3wt% (red), and at 1,000 rpm with 5 wt% (blue).

The measured transmittance spectra show the PVA polymer is nearly transparent in visible frequency. Significantly, the difference in transmittance in the PVA film thickness variation. It

originates from a degree of surface roughness, which promotes higher scattering through the increased granules¹. In other words, it elevates the reflectance, thereby diminishing the transmittance. The surface uniformity photograph of PVA films depends on the relation between the spin-curve and concentration will be discussed in Supplementary Note 6”

c)

We thank the reviewer for their comment. Indeed, we did not include the exact ellipsometry data in the original manuscript, and we agree that this detailed information could be useful for readers. The refractive index and thickness of the PVA films are characterized using commercial ellipsometry (J.A. Woollam, M-2000D). During the measurement, a deuterium/halogen lamp light source is illuminated at an incident angle of 65 degrees to the sample. The Cauchy model was then used to determine the refractive index of the PVA films. This model is the standard choice for measuring visibly transparent polymers. In the revised manuscript, we have added the specific measurement data, including the light source, incident angle, and calculation model as follows:

In the manuscript:

Line 355: “A deuterium/halogen light source was illuminated onto the sample at an incident angle of 65°, and as standard for visibly transparent polymers, the Cauchy model was used to determine the refractive index of the PVA films.”

d)

We thank the reviewer for pointing out our inconsistency for the RH. We have fixed the RH to be 80% as we intended, as well as included them in the Figures where relevant for the ease of the readers. The edited parts of the manuscript and edited Figures are as follows:

In the manuscript:

Line 102: “The measured swelling for the film spin-coated at 1,000 rpm reaches ~35.5% increase with a dramatic increase in thickness from RH 25% to 65%, and negligible change between RH 65% and 80%.”

Line 111: “Furthermore, using effective medium theory, we calculate the refractive index of PVA at RH 80% (Supplementary Note 5). Since the n of water is lower than that of PVA, the n of the swollen PVA at RH 80% slightly decreases.”

Line 372: *“The switching of imaging, which occurs by the swelling/deswelling of the PVA metasurface was captured in real-time during the reversible transition of humidity between low (RH 45%) and high (RH 80%).”*

Figure 3:

Figure 4:

Figure. 5:

Comment 3:

a) To numerically examine the effect of water molecules' absorption on the deformation of PVA, the authors performed the thermal expansion simulation of the nanostructured PVA film using commercially available software, COMSOL Multiphysics. Is there a special reason why the swelling property of PVA by humidity is analyzed in terms of thermal expansion?

b) What is the value of the relative humidity when the human breath is applied?

Response 3:

We thank the reviewer for their critical comments that have helped to improve our work regarding deformation of the PVA meta-atoms and to give some scientific background to the value of relative humidity under exhaled human breath.

a)

We chose to model the swelling properties of PVA in terms of thermal expansion under the assumption that the expansion of the PVA is isotropic. Therefore, it can be described by the equation: $\epsilon_{th} = \alpha_{th}(T-T_{ref})$, where ϵ_{th} is the thermal strain; α_{th} is the thermal expansion coefficient; and T and T_{ref} are the temperature of the material and reference, respectively. In the original manuscript, we used the isotropic thermal expansion coefficient, $\alpha_{th} = \begin{bmatrix} \alpha_{11} & 0 & 0 \\ 0 & \alpha_{11} & 0 \\ 0 & 0 & \alpha_{11} \end{bmatrix}$. This isotropic thermal expansion is the same as the isotropic expansion of hygroscopic swelling, which can be expressed as $\epsilon_{hy} = \beta_{hy}M_m(c-c_{ref})$, where ϵ_{hy} is the hygroscopic strain; β_{hy} is the hygroscopic swelling coefficient; M_m is the molar mass; and c and c_{ref} are the atmospheric and reference concentrations, respectively [R1, R2]. For isotropic expansion, these two equations are analogous, however, as the reviewer points out, this could be potentially confusing for readers in our work since the deformation is due to humidity rather than heat. Therefore, we conducted the same simulations using the hygroscopic model to give a clearer view of the expansion that is related to our experiment. As can be seen in Figure R1, the two simulations show the exact same swelling responses, justifying our original method. However, the hygroscopic model is definitely more appropriate for this work. Therefore, we have edited the swelling simulations in Figure S9 to the hygroscopic model and have amended the explanations accordingly.

a. Thermal expansion model

b. Hygroscopic swelling model

Figure R1. Comparison between (a) thermal expansion, and (b) hygroscopic model. Both show the same displacement as thermal expansion and the hygroscopic model have the same governing equation.

We have copied the edited part of the Supplementary Information below for the ease of the reviewer:

In the Supplementary Information:

“Supplementary Note 12: PVA swelling simulations

Figure S9. Numerical simulations of the swelling of Meta A. Swelling simulation of Meta A when the origin volume is swollen by (i) 110%, (ii) 120%, and (iii) 135.5%.

To simulate the swelling of PVA nanostructures with regards to the RH, we model nine PVA structures on a substrate. The size of the structures is set to $350 \times 200 \times 500 \text{ nm}^3$, to represent Meta A, and the rotation angles of blocks are set to arbitrary values between $-90^\circ \sim 90^\circ$.

The 3D swelling configuration of modeled structures is numerically calculated with the commercially available software, COMSOL Multiphysics version 5.6, using the solid mechanics and heat transfer modules. We conduct hygroscopic swelling simulations on the structures. The substrate is considered to be rigid and the centers of the bottom of the PVA structures are fixed to the substrate. We assume that the material has an isotropic coefficient of hygroscopic expansion β [m^3/kg], and β is increased from 0 until the volume swelling ratio (V/V_0) reaches 135%.” The maximum displacement is observed at the top corners of the meta-atoms, which means that any aggregation of the meta-atoms during swelling starts from the top of the meta-atoms, effectively destroying the structures.”

[R1] *Appl. Surf. Sci.* **577**, 151895, (2022)

[R2] *Sens. Actuators B*, **255**, 1343-1353, (2018)

b)

We thank the reviewer for bringing up this important point. In the original manuscript we did not confirm the relative humidity when a human breath is applied. To clarify this, we have conducted further experiments to measure the relative humidity under the influence of an exhaled human breath. To do this, we measured the relative humidity using a commercial humidity meter (Testo, Testo 625). 2 different subjects each breathed on the sensor 10 times. Each breath lasted for around 1.5 seconds, at a distance of around 2.5 cm from the sensor,

which is consistent with the experimental method when breathing on the metasurface sample that is placed on the optical microscope. The results are copied below, and we have added them to both manuscript and Supplementary Information as follows:

In the manuscript:

Line 204: “The high humidity conditions can also be purposefully produced using the breath of a human⁴⁹ (Supplementary Note 14), allowing for sensitive information to be destroyed at will after it has been retrieved by the receiver.”

Line 370: “To demonstrate the dynamic tuning process, human breath was directly exhaled onto the sample at a distance of around 2.5 cm for around 1.5 sec per breath.”

In the Supplementary Information:

“Supplementary Note 14: Measured relative humidity of human breath

Figure S12. Measured RH of human breath. The RH of the room was measured to be 45%. Two different subjects breathed on the humidity sensor at a distance of 2.5 cm away for around 1.5 sec. The dots represent the maximum values of RH measured for each breath.

To confirm the value of RH that is achieved under the exhalation of a human breath, we measured the maximum RH using a commercial humidity meter (Testo, Testo 625). Two different researchers breathed on the meter at a distance of 2.5 cm away for around 1.5 sec, 10 times each, and the maximum RH reached was recorded (Figure S12). The RH of the room in which the experiment was conducted was measured to be 45%. The mean RH for subject 1 was measured to be 80.0%, while subject 2 was 83.9%. The measured values are comparable to the values quoted in the literature (85%)⁵.”

Comment 4:

a) It is quite impressive to use the variation in optical phenomena caused by the humidity sensitivity of PVA for security label applications. However, it would be considered to be quite vulnerable to keep the original status in areas with frequent rains or high humidity. Is there a way to supplement the proposed technology for this?

b) Polymers are relatively weak to heat. How is the thermal stability of the proposed labels?

c) How many times are the reversible and repeatable process of the label kept?

Response 4:

We thank the reviewer for their instructive comments about the repeatability and lifetime of our PVA-based metasurfaces. We have conducted further experiments to confirm such properties, which has helped to improve our work further. The specific answers to each comment are below.

a)

We thank the reviewer for their important comment. As the reviewer points out, such PVA-based metasurfaces could be vulnerable to areas with frequent rain, as water would be able to wash away the PVA completely. This could be protected using simple strategies such as covering the metasurfaces to block any direct interaction with rain water, however, this would result in a large trade-off in terms of response time, as the surface area that can absorb the H₂O in the air would be reduced. In terms of locations with generally high humidity, the metasurfaces could be designed to work in reverse, where they are fabricated in high humidity conditions and rather than swelling to reveal hidden information, the deswelling process could be used as an alternative. To reflect these important points, we have added the following to the conclusion:

Line 301: "The proposed PVA metasurfaces could be vulnerable to harsh weather conditions such as frequent rain and locations with generally high RH. This could be supplemented with protective materials to block direct contact with water, with a trade-off in terms of response time. For locations with constant high humidity, the metasurfaces could be designed with the baseline RH of the specific place in mind and adapted accordingly, while other hydrogels could also be explored."

b)

We thank the reviewer for their important comment about the thermal stability of our PVA metasurfaces. To confirm the stability for real-life applications, we conducted additional

experiments to assess the thermal response of the PVA metasurfaces. We have added the following to the revised manuscript and supplementary information:

In the manuscript:

Line 268: “This robust, reversible, and repeatable response allows the hidden information in the reflected visible color print to be decrypted at will, which has potential uses for secure information storage. Additionally, we prove the proposed PVA metasurfaces are robust even after exposure to thermal stress of up to 70 °C, the process is maintained for 100 exposures to 80% RH (Supplementary Note 19).”

In the Supplementary Information:

“Supplementary Note 19: Robustness of the PVA metasurfaces

Figure S17. Robustness of the PVA metasurfaces. (a) SEM image of thermally annealed reversible PVA metasurfaces after exposure to 70 °C for 72 h. Inset: Holographic image. (b) SEM images of the annealed sample after (i) 20, and (ii) 100 exposures to 80% RH. Insets: Holographic images. Scale bars: 1 μm

We conducted experiments to confirm the robustness of the PVA metasurfaces in response to exposure to high temperatures by baking them at 70 °C for 72 h. From the scanning electron microscope (SEM) images (Figure S17), it is clear to see that the meta-atoms geometries are maintained without any distinguishable defects. This is further confirmed through the holographic image, which is reproduced successfully after the exposure to high temperatures. Furthermore, we confirm the performance of the -annealed sample by exposing it to 1.5 sec of exhaled breath (RH 80%) 100 times. The SEM images after 20 and 100 exposures show no obvious defects and the holographic images are reproduced successfully, proving the potential for the use of our PVA metasurfaces in everyday room conditions which have temperatures and RH that are generally lower than 70 °C and 80%, respectively.”

c)

We thank the reviewer for their important comment about the repeatability of our reversible PVA metasurfaces. To verify the repeatability of our PVA metasurfaces we expose the sample to exhaled breath for 100 repetitions. The reflected image from the optical microscope, SEM image of the metasurface, and holographic images after 0, 20, 40, and 100 repetitions were recorded and supplied below. We have also updated Figure 5 of the main manuscript to reflect this comment as follows:

In the manuscript:

Line 260: “Thanks to the Pt layer, on the contrary to the previous examples, the meta-atoms preserve their nanoscale geometry after exposure to RH 80%. This allows for the holographic image to be displayed again, and for the color print to be encrypted when the RH is reduced back to room level. The decryption is experimentally measured to occur when the RH is around 68% (Supplementary Note 18). The robustness of this reversibility is confirmed by exposing the PVA metasurface to exhaled breath up to 100 times (Fig. 5c). The SEM images show that the meta-atoms morphologies are maintained, which allows for the repeatable decryption of the hidden color image and the recreation of the multiplexed metahologram. Videos of these two reversible processes are provided in Supplementary Video 3 and Supplementary Video 4.”

“Figure 5. Reversible PVA metasurfaces for a two-channel security platform. (a) Schematic illustration of a Pt coated PVA meta-atom on top of a-Si:H. (b) Photographs of the holographic images under illumination of 532 nm coherent light, and reflected optical microscope images under white light illumination of the Pt coated PVA metasurfaces over 100 cycles of RH 80% exposure using human breath. Scale bars: 50 μm . (c) SEM images of the PVA metasurface (i) as fabricated, (ii) after 5 and (iii) 100 cycles of exposure to RH 80%. The nanostructures are maintained, allowing for the holographic information to be recovered. Scale bars: 1 μm .”

Comment 5:

a) When a platinum (Pt) thin film is deposited on the surface of PVA films, what is their transmittance? Is there an optical absorption (or plasmonic effect) by the Pt thin film?

b) Continued with the comment #3-2), can the relationship between relative humidity and image variation be quantified in the repeated process? This can provide the threshold of the relative humidity for the decryption of encrypted information.

Response 5:

We appreciate the important suggestions for improving our work regarding the optical properties and robustness of our reversible PVA metasurfaces. The replies to each comment are below:

a)

In section ‘2.3.3. Reversible two-channel humidity sensitive optical security’ we added the Pt coating to our metasurfaces as a way to preserve the nanostructures after exposure to humid conditions, but as the reviewer correctly points out, we did not give any characterization of the metasurfaces with this added layer. To that note, we have conducted additional experiments and have included the results below. Furthermore, we have added the following to the manuscript and Supplementary Information:

In the manuscript:

Line 249: “This allows the individual meta-atoms to retain their nanoscale structures after the swelling and deswelling process, so both the encrypted print and holographic information can be retained. The optical properties of a Pt metasurface made up of Meta A structures are measured to confirm the existence of any additional effects from the Pt layer (Supplementary Note 16).”

In the Supplementary Information:

“Supplementary Note 16: Measured absorption characteristics

Figure S14. The measured transmittance, reflectance, and absorptance spectra of Meta A. Measured spectra for (a) PVA metasurfaces made up of Meta A, and (b) the same metasurfaces with a 5 nm Pt layer. Transmittance: black, reflectance: red, and absorptance: blue.

To determine the effect of adding the Pt coating to our PVA metasurfaces, we prepared metasurfaces made up Meta A and measured transmittance (T) and reflectance (R), and calculated the absorption (A) with and without the platinum (Pt) coating. We measure T and R of the uncoated PVA metasurface using spectrometry (Figure S14a). After that, an ultrathin, ~ 5 nm, Pt film is coated using an ion sputter at 20 mA for 60 [S6], and the T and R are then measured again., A is calculated using $A=1-R-T$. The Pt coating causes around a 20-30% of extra absorption due to the high extinction coefficient of Pt in the visible regime, however, no plasmonic resonances are excited.”

b)

We thank the reviewer for recommending the clear description of relation between relative humidity and image decryption. To quantify the threshold of the decryption points with the reversible Pt-coated PVA metasurfaces, we obtained images under varied humidity conditions using optical microscopy. A humidity sensor was integrated to quantify the relative humidity while simultaneously obtaining the images (Figure R2.a). The relative humidity was gradually increased from RH 58% to 82% by blowing N_2 through water vapor into the inlet. The color print images are decrypted at RH 68% (Figure R2.b). Considering that the measured RH of human breath is over 80% as we presented in response to Comment 3b, the quantified threshold (of RH 68%) validates our encryption/decryption system using human breath. We have added the detailed explanation to both revised manuscript and Supplementary Information.

Figure R2. Measurement of the decryption threshold. (a) Measurement setup to quantify threshold of decryption. The chamber is uncovered in the figure for clarity, but it is covered in the experiment. (b) Optical images are obtained at i) RH 58%, ii) RH 68%, and iii) RH 82%. Scale bar: 50 μm .

In the manuscript:

Line 260: “Thanks to the Pt layer, on the contrary to the previous examples, the meta-atoms preserve their nanoscale geometry after exposure to RH 80%. This allows for the holographic image to be displayed again, and for the color print to be encrypted when the RH is reduced back to room level. The decryption is experimentally measured to occur when the RH is around 68% (Supplementary Note 18). The robustness of this reversibility is confirmed by exposing the PVA metasurface to exhaled breath up to 100 times (Fig. 5c). The SEM images show that the meta-atoms morphologies are maintained, which allows for the repeatable decryption of the hidden color image and the recreation of the multiplexed metahologram. Videos of these two reversible processes are provided in Supplementary Video 3 and Supplementary Video 4.”

In the Supplementary Information:

“Supplementary Note 18: Measurement of the decryption threshold for the Pt coated PVA metasurfaces.

Figure S16. Measurement of the decryption threshold. (a) Measurement setup to quantify threshold of decryption. The chamber is uncovered in the figure for clarity, but it is covered in the experiment. (b) Optical images are obtained at i) RH 58%, ii) RH 68%, and iii) RH 82%. Scale bar: 50 μm .

The threshold RH to decrypt the hidden structural color image in the Pt-coated metasurfaces is measured using a humidity sensor implanted optical microscopy setup (Figure S16a). The sample had already been exposed to human breath 100 times to validate the repeatability of the optical encryption system. The RH is increased in the chamber by inserting N_2 gas to gradually increase the RH from 58% to 82%. The image is decrypted at RH 68%, and continues to be visible until around RH 82%, which correlates to the RH of human breath.”

Comment 6:

Can the authors show the biological safety of the proposed final PVA labels to confirm their low toxicity and biocompatibility, **not** referring to the previous literature? In the fabrication process, some toxic chemicals including silane coupling agent (Trichloro-1H, 1H, 2H, 2H-Perfluorooctyl-silane) and toluene are used.

Response 6:

We thank the reviewer for bringing up this extremely important point about the toxicity of our PVA metasurfaces. As the reviewer points out, toxic chemicals (e.g. silane agent, etc.) are used to adjust surface energy of soft-mold for easy detach from structured PVA. However, as shown in Figure S9, no toxic chemical is left in the final PVA metasurfaces. To clear up these points, we have conducted additional experiments to prove the absence of the silane coupling agent in the final samples. With regards to toluene, it is an extremely volatile substance that almost instantly evaporates into the surroundings, so cannot be present after the PVA metasurfaces have been printed. Since the silane agent has toxic elements such as F and Cl that are not present in the chemical structure of PVA. Therefore, through energy-dispersive X-ray spectroscopy (EDS) analysis, we are able to identify the amount of each element in both the soft mold and the final one-step imprinted PVA metasurface. This has been addressed in the revised manuscript as follows:

In the manuscript:

Line 154: “Furthermore, we experimentally confirm that the potentially harmful coupling agents used in the fabrication of the soft mold do not exist in the final PVA metasurfaces to prove their potential for such implementations (Supplementary Note 9).”

In the Supplementary Information:

“Supplementary Note 9: Experimental confirmation of the non-toxicity of the PVA metasurfaces

Figure S7. Experimental verification of the non-toxicity of the PVA metasurfaces. (a) The chemical structures of (i) polyvinyl alcohol (PVA) and (ii) the silane agent (Trichloro-1H, 1H, 2H, 2H-Perfluorooctyl-silane). (b) Energy-dispersive X-ray spectroscopy (EDS) analysis of the (i) soft mold, and (ii) one-step imprinted PVA metasurface. Scale bars: 500 nm.

Since there are a few potentially harmful substances used in the fabrication of the soft mold that is used to replicate the PVA metasurfaces, we confirm that there is no trace of them left in the final one-step imprinted metasurfaces. First, it is worth mentioning that one of the coupling agents used to create the soft mold, toluene, is extremely volatile. This means that it almost instantly evaporates into the surrounding air, and therefore cannot be found in the final PVA metasurface. Another important substance that could be harmful to humans is silane coupling agent (Trichloro-1H, 1H, 2H, 2H-Perfluorooctyl-silane). The chemical structures of PVA and silane coupling agent are shown in Figure S7a. The elements fluoride (F) and chlorine (Cl) are only found in the harmful agent, therefore, to prove the absence of such toxic substances, we conduct energy-dispersive X-ray spectroscopy (EDS) analysis on both the soft mold and the one-step imprinted PVA Metasurfaces (Figure S7b). The results clearly show that the harmful elements are not transferred to the final PVA metasurfaces as there is no trace of them, proving their non-toxicity and potential for use in the food and biomedical industries.”

Comment 7:

The sizes of multiplexed structural coloration and holography labels are in the range of a number or tens of a micron. In addition, most images are measured by an optical microscopy system. Such a bulky reading system would be a limitation of the proposed technology in practical applications. Can the labels be further scaled up or be read by easy-accessible readers (e.g., smartphones)?

Response 7:

We appreciate the constructive and helpful suggestions for practical applications of the proposed PVA metasurfaces. Considering the time-consuming and high-cost fabrication process of the commercial electron beam lithography, we selected reasonably sized metasurfaces of $250 \times 250 \mu\text{m}^2$, which allows us to easily observe the structural coloration using an optical microscopy system, as well as being big enough to measure the optical properties using spectroscopy. We agree with the comment of the reviewer in that such a bulky reading system would be a huge limitation for practical applications. However, our demonstration was more of a first proof-of-concept at the lab scale, and such size limitations can be easily overcome by using larger master molds, and soft molds, and in the future, roll-to-roll nanoimprint technology could open up the path for continuous printing of such metasurfaces. Considering that cm-scale metasurfaces have been currently reported with electron-beam lithography processes [R3] and UV lithography [R4, 5], we believe that our system can be implanted on easy-accessible smartphone readers when the master mold sizes are scaled-up. In Figure 2c we included an example of large-scale fabrication on flexible substrates to prove the feasibility of the nanoimprinting processes, however the master mold was made using photolithography, which limits the possibility of creating nanoscale structures. To clear up this critical point, we have added the following to the manuscript:

In the manuscript:

Line 291: *“Although the structural color of the metasurfaces in this work require optical microscopy systems to image, this is not a fundamental limitation of the design, which can be simply scaled-up based on the size of the master mold. By implementing fabrication methods to create centimeter-scale metasurfaces for the master mold, the encrypted color images in the PVA metasurfaces proposed here could easily be read by eye, or using everyday smartphone readers. Furthermore, the one-step nanoimprinting method leads well for potentially extremely high-throughput techniques such as roll-to-roll imprinting.”*

[R3] Nat. Commun. 9, 4562, (2018)

[R4] Nat. Commun. 13, 2409, (2022)

[R5] Nano Lett. 19, 8673-8682, (2019)

Comment 8:

a) *Please double check the scale bars or values of the scale bar in all figures and figure captions. In particular, the scale bars in Fig. 5c should be smaller than 50 μm .*

b) *Please provide the specific values of the high RH concentrations in the figure captions and manuscript.*

c) *In Table S1 of Supplementary Note 2, please provide the standard deviation of measured thickness for spin-coated PVA films.*

d) *In Supplementary Note 7, please check the value of $\sim 96\%$ in the reflectance spectra of a bare SiO_2 substrate. It seems to be $\sim 4\%$.*

e) *In Supplementary Note 7, what are the evident peaks in the reflectance spectra for the nanocuboid and nanocylinder with width or diameter values larger than 200 nm?*

f) *Overall, there are mistakes on grammar and spelling in the manuscript. Fine/minor spell check in the English language and style should be required.*

Response 8:

We thank the reviewer for pointing out these mistakes. We have carefully proof-read our manuscript a number of times to clear up any pointed-out minor issues. All revisions have been copied below for the ease of the reviewer.

a)

Line 545: “**Figure 2. Fabrication of PVA nanostructures using NIL.** (a) The fabrication process for PVA NIL. (i) The master mold is fabricated using EBL and (ii) a negative soft mold is casted using PDMS. The soft mold is then coated with PVA, and the NIL process is conducted to produce (iii) PVA metasurfaces. Scale bars: 1 μm (b) SEM images of (i) 1D gratings, and (ii) 2D arrays of nanostructures and (iii) nanoholes fabricated using PVA NIL. Scale bars: (i), (ii), (iii) 1 μm , and (iv), (v), (vi) 500 nm.”

Line 555: “**Figure 3. PVA metasurfaces for multiplexed structural color and metaholography for humidity sensitive smart labels.** (a) Schematic illustration of the PVA meta-atom for humidity sensitive smart labels. (b) SEM images of the PVA metasurface (i) before and (ii) after exposure to RH 80% conditions. Scale bars: (i) 1 μm , (ii) 100 nm. (c) Calculated cross-polarization efficiencies at the working wavelengths of (i) 635, (ii) 532 and (iii) 450 nm. Two meta-atoms (denoted as Meta A and Meta B) are selected. P and H are fixed at 400 nm and 500 nm. (d) Reflected optical microscope image of the metasurface under white light (i) before and (ii) after exposure to RH 80% conditions. Scale bars: (i), (ii) 100 μm .”

Line 581: “**Figure 5. Reversible PVA metasurfaces for a two-channel security platform.** (a) Schematic illustration of a Pt coated PVA meta-atom on top of a-Si:H. (b) Photographs of the holographic images under illumination of 532 nm coherent light, and reflected optical microscope images under white light illumination of the Pt coated PVA metasurfaces over 100 cycles of RH 80% exposure using human breath. Scale bars: 50 μm . (c) SEM images of the PVA metasurface (i) as fabricated, (ii) after 5 and (iii) 100 cycles of exposure to RH 80%. The nanostructures are maintained, allowing for the holographic information to be recovered. Scale bars: 1 μm .”

Also, we added a description for sufficient understanding as follows:

In the manuscript:

Line 349: “Pt-coated for reversible PVA metasurface: The top ultrathin (5 nm) Platinum (Pt) was deposited by commercial Pt coater (Hitachi, MCI 1000). The deposition process was conducted in the 20 mA for 60 s for forming a 5 nm ultrathin film onto the PVA metasurface.”

b)

We added all of the measured relative humidity values into both the figures and manuscript as follows:

In the manuscript:

Line 189: “After the metasurfaces are exposed to RH 80%, the meta-atoms expand and aggregate, resulting in the destruction of the nanostructures.”

Line 203: “After undesired exposure to RH 80% humidity conditions, both pieces of information are lost, informing the user that the product has been compromised.”

Line 212: “We use this to design a highly secure two-channel encryption system by exploiting the fact that the geometry of the meta-atoms is destroyed after exposure to RH 80% conditions.”

Line 218: “At RH 45%, a single uniform color is displayed in reflection, hiding the encrypted information, until it is irreversibly revealed through exposure to RH 80% conditions.”

Line 228: “Through exposure to RH 80% conditions using a human breath, since the swelling rate of Meta A and Meta B differ due to their overall volumes, the locations of the two different meta-atoms become clearly visible to uncover the hidden reflected image (Fig. 4d).”

Line 235: “As can be seen in the video, the hidden image is vaguely visible after the first two breaths, and completely destroyed after that, while the holographic information instantly disappears after exposure to RH 80% conditions.”

Line 257: “When exposed to RH 80% the hidden word 'PASSWORD' is decrypted temporarily due to the difference in swelling of Meta A and Meta B, while the holographic information is temporarily hidden.”

Line 260: “Thanks to the Pt layer, on the contrary to the previous examples, the meta-atoms preserve their nanoscale geometry after exposure to RH 80%. This allows for the holographic image to be displayed again, and for the color print to be encrypted when the RH is reduced back to room level. The decryption is experimentally measured to occur when the RH is around 68% (Supplementary Note 18).”

Figure 3:

Figure 3. PVA metasurfaces for multiplexed structural color and metaholography for humidity sensitive smart labels. (a) Schematic illustration of the PVA meta-atom for humidity sensitive smart labels. (b) SEM images of the PVA metasurface (i) before and (ii) after exposure to RH 80% conditions. Scale bars: (i) 1 μm , (ii) 100 nm. (c) Calculated cross-polarization efficiencies at the working wavelengths of (i) 635, (ii) 532 and (iii) 450 nm. Two meta-atoms (denoted as Meta A and Meta B) are selected. P and H are fixed at 400 nm and 500 nm. (d) Reflected optical microscope image of the metasurface under white light (i) before and (ii) after exposure to RH 80% conditions. Scale bars: (i), (ii) 100 μm . (e) Photographs of the holographic images produced from the PVA metasurface when illuminated with red, green, and blue lasers with wavelengths of 635, 532, and 450 nm, respectively, (i) before and (ii) after exposure to RH 80% conditions. Both the nanoprint and holographic information are completely destroyed and unrecoverable after exposure to RH 80%.

Figure 4:

Figure 4. Irreversible PVA metasurfaces for a single-use two-channel security platform. (a) Schematic illustration of a PVA meta-atom on top of a-Si:H (b) SEM images of the PVA metasurface (i) before and (ii) after exposure to RH 80% conditions. Meta A and Meta B are reproduced with high fidelity using NIL. Scale bars: (i) Top: 500 nm, Bottom: 100 nm. (ii) 500 nm. (c) Photographs of the holographic images produced from the PVA metasurface when illuminated with red, green, and blue lasers with wavelengths of 635, 532, and 450 nm, respectively, (i) before and (ii) after exposure to RH 80% conditions. The hologram is instantaneously destroyed. (d) Reflected microscope images of the PVA metasurface (i) before, (ii) during, and (iii) after exposure to RH 80% conditions. The encrypted print is uncovered as the meta-atoms swell at different rates. Scale bar: 50 μm .

Figure 5:

Figure 5. Reversible PVA metasurfaces for a two-channel security platform. (a) Schematic illustration of a Pt coated PVA meta-atom on top of a-Si:H. (b) Photographs of the holographic images under illumination of 532 nm coherent light, and reflected optical microscope images under white light illumination of the Pt coated PVA metasurfaces over 100 cycles of RH 80% exposure using human breath. Scale bars: 50 μm . (c) SEM images of the PVA metasurface (i) as fabricated, (ii) after 5 and (iii) 100 cycles of exposure to RH 80%. The nanostructures are maintained, allowing for the holographic information to be recovered. Scale bars: 1 μm .

c)

We have modified Figure S1 to include the standard deviation of the measured PVA thickness when it is spin coated at the identical conditions 5 times. The modified Figure S1 shows more clear description of the spin-coating curve. Below is the modified Figure S1 copied for the ease of the reviewer:

In the Supplementary Information:

“Supplementary Note 1: Thickness of spin-coated 3wt% and 5wt% PVA thin films

Figure S1. Measured spin-coated PVA film thickness. The measured thickness of PVA films spin-coated from 1,000 rpm to 4,000 rpm for PVA concentrations of 3wt% (black) and 5wt% (red). Error bars: one standard deviation.”

d)

We thank the reviewer for carefully reading our manuscript and pointing out this important mistake. We have verified and corrected it as follows:

In Supplementary Note 10:

“We verify different reflectance spectra can be obtained by changing filling ratio and geometry of subwavelength nanostructured PVA. When the filling ratio is close to zero, its reflectance spectra is close to bare SiO₂ substrate (~4%).”

e)

We thank the reviewer for their important question. We attribute the peaks in the spectra to arise from the standard reflection from the SiO₂ substrate, however, when the size of the

nanostructure becomes large enough, the increase in the effective refractive index in the meta-atom unit cell satisfies impedance matching conditions for antireflection at certain wavelengths. This creates the dips in the reflectance spectra. We have added the following to clear this up:

In Supplementary Note 10:

“However, when the width of the nanocuboid or diameter of nanocylinder are larger than 200 nm, evident peaks appear due to impedance matching that satisfy antireflection conditions at certain wavelengths.”

f)

We thank the reviewer for their useful comment to help improve the English presentation of our manuscript. We have carefully proof-read our manuscript a number of times to clear up any confusing sentences.

Reviewer 2

In this manuscript entitled “Tunable metasurfaces via the humidity responsive swelling of single-step imprinted polyvinyl alcohol nanostructures”, the authors have experimentally demonstrated tunable polyvinyl alcohol (PVA) metasurfaces by using the nanoimprinting technique. Owing to the humidity-sensitivity of hydrogel, the PVA nanostructures swell and aggregate under the high relative humidity (RH) conditions, and lead to optical information destruction. Furthermore, by depositing 10 nm Pt onto the PVA metasurface, the top metal preserves the PVA nanostructures from being completely destroyed under the high RH and therefore provides a reversible strategy for two-channel optical information switching.

However, after a careful review of this work, I think that the demonstration of this work is more like a progressive work from their recent publication in Science Advances (Jung, C., Kim, S.J., Jang, J., Ko, J.H., Kim, D., Ko, B., Song, Y.M., Hong, S.H. and Rho, J., 2022. Disordered-nanoparticle-based etalon for ultrafast humidity-responsive colourimetric sensors and anti-counterfeiting displays. Science advances, 8(10), eabm8598). The departure from that work is not big enough and makes this submission lack the novelty and impact to be suitable for publication in Nature communications. The proposed uses of PVA metasurface, such as single-use smart labels and irreversible optical security, are designed by a conventional method and have been demonstrated before similarly. Additionally, in my humble opinion, the PVA metasurface demonstrated in this manuscript is hard to be applied in daily life due to the uncontrollable ambient humidity. When the ambient humidity around the sample is higher than the damage threshold, the encrypted optical information might be automatically destroyed, thus making manual decryption impossible. In addition, the author discussed quite in detail about fabrication processing of PVA films and patterns, therefore I think the paper is rather more appropriate to narrow audiences and is more suitable for specific topics in some material-focused journals.

Here are some further comments in detail for improvement or correction.

Response:

First of all, we would like to give our appreciation to the reviewer for carefully evaluating our manuscript and providing us with such important and critical comments that have helped to further improve our manuscript. We are confident that with the revisions and additions that we have made, the quality and scientific content of our work has drastically improved, and we hope that we can convince the reviewer to support our manuscript for publication in Nature Communications.

Before providing our responses to the comments for improvement and correction that were kindly given by the reviewer, we will first try to address the main concerns that were brought up, to hopefully demonstrate why our work is of high enough novelty and impact, as well as being interesting for the broad readership of Nature Communications.

First, the reviewer stated their concern that the departure from the work in reference 26 is not big enough, and therefore lacks novelty and impact. We are extremely grateful that editor has

allowed us the chance to outline the advantages and differences of our work, and justify the functionality of our work compared to reference 26.

The work in question developed humidity sensors using a different hydrogel, chitosan, as a functional layer in a classic Fabry-Perot type design. The main focus is on the drastically improved response time of the device that was achieved by using disordered metallic nanoparticles rather than a solid deposited film of metal as the top layer of the Fabry-Perot resonator. The color QR codes in that work were fabricated using a multistep photolithography for 2-10 μm sized pixels.

We believe that our work here, on the other hand, is based on completely different concepts. Our metasurfaces are fabricated using one-step nanoimprint lithography (NIL). To the best of our knowledge, this is the first demonstration of such nanoimprint lithography using hydrogels. Although, some dielectric-nanoparticle embedded polymer-based nanostructures have been proposed [R1], they provide no method of tunability. Whereas the use of hydrogels such as PVA allow for the realization of tunable, reversible, and irreversible nanophotonic applications. This is somewhat related to the final concern of the reviewer that this work would be suited to a journal with a narrower audience. We feel that it is important, as the first example of NIL with PVA, to demonstrate the various potentials of using PVA with NIL to produce large scale films that can be directly printed on any substrate, including flexible polymers that can be attached to bottles or surfaces of any shape. Furthermore, we expect that this work could also act as a guide for future work on using different hydrogels or materials for NIL.

The utilization of NIL with PVA allows us to exploit the full range of manipulation of the optical properties of light using nanostructures, with a tunable response that is allowed by the PVA. This means that we are able to use a periodicity of just 400 nm for our unit cell, providing ultra-high resolution for structural color images while also multiplexing metaholography, which we achieve by designing 2 distinct meta-atoms that provide similar reflection properties under certain RH conditions, and different properties when they swell at different rates when exposed to higher RH. We exploit this idea to numerically design and experimentally demonstrate 3 different devices. The first displays a publicly available color print, with an additional channel that includes a holographic image, which are both destroyed when the PVA is exposed to high humidity conditions, such as under the breath of a human. We envision that this device has potential uses as smart labels for the food and medical industries, since the PVA is non-toxic, and the metasurfaces can be easily printed and attached to packaging or bottles. The second device encrypts the color print, while displaying a holographic image. To decrypt the color print image, the PVA metasurface must be exposed to the correct humidity conditions, which in turn destroys both the color print and holographic information stored in the metasurface. We believe that this device could be useful in applications where the end user needs to hide sensitive information in the color print, and can be made aware that their information has been compromised by using the existence or absence of the holographic information as a secret key. Finally, by adding a thin platinum coating layer, we demonstrate a reversible version of our multiplexed PVA

metasurfaces. The Pt allows the meta-atoms to maintain their nanoscale geometries even after exposure to high humidity conditions. This has potential for robust applications in various optically variable devices for optical security and encryption, and to the best of our knowledge, has never been demonstrated before.

The reviewer comments that our work is designed using conventional methods, and has been demonstrated before similarly. To the best of our knowledge, the nanostructuring of PVA or hydrogels for multiplexed structural color and metaholography has not been demonstrated before. We are aware of various work that, similarly to reference 26, used the Fabry-Perot type design to create tunable structural colors, such as ref [R2]. Ref [R3] demonstrates multiplexed color and holography, but again through stepwise photolithography in the Fabry-Perot setup. Furthermore, the design of ref [R3] only shows the designed highly pixelated color images under the illumination of certain wavelengths of light. Our designs, however, are displayed under the illumination of white light, which means they have the potential to be read simply by the naked eye, or using a readily available smart phone camera. Additionally, the pitch of a single super-pixel is 4 μm , compared to the nanoscale resolution that we are able to provide using NIL.

Finally, we wholeheartedly agree with the concerns regarding the application of our PVA metasurfaces in daily life due to the uncontrollable ambient humidity. However, as is the case with most research at the lab-scale, our work is more of a proof-of-concept idea, rather than a final product that could be directly implemented into real-world applications. We would also argue that most applications that we would aim to target with these PVA metasurfaces would necessarily already be fairly protected, in terms of food being refrigerated and medical supplies generally being protected from unwanted contamination by being stored carefully and cleanly, where any discrepancies could be easily detected using our PVA metasurfaces. That being said, for locations that are unavoidably exposed to the elements and have the potential for being directly impacted by rain, we envision a form of physical protective barrier would need to be implemented in order to protect the metasurfaces. However, this would compromise the responsivity, as the water molecules should first penetrate the protective barrier. We have included a discussion of this important concern in the conclusion of the manuscript as follows:

In the manuscript:

Line 301: “The proposed PVA metasurfaces could be vulnerable to harsh weather conditions such as frequent rain and locations with generally high RH. This could be supplemented with protective materials to block direct contact with water, with a trade-off in terms of response time. For locations with constant high humidity, the metasurfaces could be designed with the baseline RH of the specific place in mind and adapted accordingly, while other hydrogels could also be explored.”

[R1] Nat. Commun., 11, 2268, 2020

[R2] Nanophotonics, 10(16), 2021

[R3] Adv. Funct. Mater.2022, 32, 2110022

Response to reviewer 2

Comment 1:

For the demonstration of large-scale fabrication, the PVA metasurface is printed on a flexible substrate, as shown in Figure 2c. The author should clarify that the structures are fabricated on the whole substrate or part of it.

Response 1:

We thank the reviewer for their crucial comment. For the flexible and large-scale fabrication, the metasurface is indeed printed across the whole substrate, but we missed to explicitly disclose this important fact. The master mold was produced using photolithography rather than electron-beam lithography, allowing us to produce a larger master mold that can be copied through NIL across large areas. This is also an important point for demonstrating the potential for upscaling the size of our metasurfaces for practical applications. Therefore, we have added the information about the large-area structures printed across the whole flexible substrate area as follows:

In the manuscript:

Line 545: “Figure 2. Fabrication of PVA nanostructures using NIL. (a) The fabrication process for PVA NIL. (i) The master mold is fabricated using EBL and (ii) a negative soft mold is casted using PDMS. The soft mold is then coated with PVA, and the NIL process is conducted to produce (iii) PVA metasurfaces. Scale bars: 1 μm (b) SEM images of (i) 1D gratings, and (ii) 2D arrays of nanostructures and (iii) nanoholes fabricated using PVA NIL. Scale bars: (i), (ii), (iii) 1 μm , and (iv), (v), (vi) 500 nm. (c) Photograph of cm-scale PVA metasurface printed on a flexible polycarbonate (PC) film using NIL. The master mold was fabricated using photolithography. Scale bar: 5 cm. (d) Photographs of the PVA metasurface attached to the curved surface of a glass bottle. Scale bar: 1 cm. Inset: Close-up of the $500 \times 500 \mu\text{m}^2$ PVA-metasurfaces. Inset scale bar: 1 mm.”

Comment 2:

In Figure 3c, the η_{cross} of Meta A and Meta B seem to locate at a similar colour level in the mapping at the wavelength of 635 nm. In fact, the η_{cross} of Meta B (4.45) is twice as the Meta A (2.01). To make the data presentation more scientific and make sense, the numerical value of the colour bar should be provided rather than normalizing it to the maximum and minimum.

Response 2:

We appreciate the helpful suggestions to improve the scientific presentation of our work.

We agree that normalized color bar of the conversion efficiency plot could cause readers to misunderstand the information. Therefore, we have renormalized the scale bars to show 0 to 10%, which we believe provides a much more scientific presentation for readers. We have included the modified part of Figure 3 below for the ease of the reviewer:

Comment 3:

In Figure 3d(i), the optical microscope image of the QR code shows that the blue area has some green components. Please comment on the colour unevenness.

Response 3:

We thank the reviewer for their important comment about the unevenness of the structural color images. To highlight this issue and our explanation, we have included the figure, along with zoomed in sections of the uneven color that the reviewer mentions (Figure R1).

Figure R1. Color unevenness of fabricated QR-code PVA metasurfaces.

We believe that this unevenness is due to the magnification of the optical microscopy setup when imaging the PVA metasurfaces as we focus on the surface, which allows us to image the near field phase modulation due to the meta-atoms. To confirm this, we investigated the phase map that was implemented for the metasurface (Figure R2). It is clear to see that the encoded phase map has the same wave-like patterns as is appear in the optical microscope images of the structural color.

Figure R2. Retrieved phase map for hologram generation. (a) Total phase map for QR-code metasurfaces, and (b) Magnified phase map images that is the red box of (a) the total phase map.

This effect is often seen in optical microscopy images of metasurfaces, and has been seen extensively in metalenses. An example of two optical microscopy images of a polarization dependent metalens is shown as in Figure R3. The imparted phase can be clearly seen.

Figure R3. Optical microscopy images of a polarization dependent metalens. Under (a) x-polarized incident light, and (b) y-polarized incident light. The imparted phase can be clearly seen.

To clarify this for readers, we have revised the manuscript and Supplementary Information as follows:

In the manuscript:

Line 195: “Although the color prints appear to show an unevenness for the same meta-atoms, this is due to the use of optical microscopy to image the color prints (Supplementary Note 13).”

In the Supplementary Information:

“Supplementary Note 13: Apparent unevenness in the reflected color

Figure S10. Apparent color unevenness of the fabricated QR-code PVA metasurfaces.

In the images of the color prints taken using optical microscopy, a slight variation in color can be seen, especially under high magnifications. This is due to focusing on the near-field of the metasurface, which allows us to see the phase modulation due to the meta-atoms. To confirm this, the implemented phase map for the metasurface is shown in Figure S11. It is clear to see that the encoded phase map has the same wave-like patterns as is appear in the optical microscope images of the structural color prints.

Figure S11. Retrieved phase map for hologram generation. (a) Total phase map for QR-code metasurfaces, and (b) Magnified phase map images that is the red box of (a) the total phase map.”

Comment 4:

Figure 5 shows the reversible holography by changing the relative humidity (RH). However, the working mechanism seems unclear. Why do the PVA structures swelling cause the holographic image to disappear? The authors should provide more explanation and discussion about the principle of this demonstration.

Response 4:

We thank the reviewer for their critical comment. After reviewing our manuscript, we agree that we did not provide enough explanation and discussion about why the swelling of the PVA meta-atoms causes the holographic images to disappear. This would be extremely important information and not obvious for any readers who are not familiar with the concepts that underpin metaholography, which is extremely important for a journal with such broad readership as Nature Communications.

The required phase for the metaholography is encoded using the concept of geometric phase, which is commonly referred to as PB phase, which allows for a desired phase to be imparted on circularly polarized light as it interacts with the metasurface. To achieve the concept of geometric phase, the conversion between the orthogonal states of circularly polarized light, i.e., left, and right circularly polarized light (LCP and RCP), is exploited. To achieved this, anisotropic meta-atoms are required, which allows for the cross-polarized component of the transmitted light to be manipulated. The value of this is defined as the cross-polarization conversion efficiency, which is defined as the amount of LCP (RCP) light that is converted to the opposite, RCP (LCP) light after interaction with the meta-atom. The meta-atoms are then rotated in-plane across the metasurface, according to the desired phase required to achieve the computer-generated hologram. The details are given in terms of the Jones vector in Supplementary Note 11.

Since the holographic image disappears under the high humidity conditions of a human breath, there could be 2 explanations. Either 1) the phase of the meta-atoms does not satisfy the required phase map anymore, possibly due to swelling, or 2) the conversion efficiency of the meta-atoms has decreased to be low enough that the holographic image cannot be distinguished from the background light. The experimental verification of this is extremely challenging, as we are not able to use the humidity chamber in the SEM to monitor the exact swelling in real-time. Therefore, we take a logical approach to conclude that we can attribute reason 2) as the factor that makes the holographic image disappear.

In more detail. We implement PB phase, which is determined only by the in-plane rotation of anisotropic meta-atoms, to achieve the desired phase map. This is extremely robust to

variations in the geometry of the meta-atoms. Therefore, we conclude that even after isotropic swelling of all the meta-atoms by up to 35%, the phase imparted on the incoming circularly polarized plane wave through the PB metasurface would not be sufficiently different from the desired one. Additionally, to completely destroy the conversion efficiency due to swelling, it would require that the PVA along the long axis should swell much more than the short axis, creating a symmetrical meta-atom shape. However, this is not possible for both of Meta A and Meta B, as they have different geometries. Through these arguments, we conclude that reason 1) cannot be the cause of the disappearing holographic image.

Moreover, thanks to the reviewer's comment, we noticed that during the exhalation of a breath onto the metasurface, the holographic image appears to show an increased background noise. We initially wrongly assumed that this is the converted light being dispersed randomly rather than to the desired holographic image. To confirm this, we used the amorphous silicon (a-Si) master mold, and treated it with hydrophobic and hydrophilic coatings. Since a-Si has no response to humidity, it can give us an insight into the effect of breathing on a metasurface in terms of the reproduced holographic image (Figure R4). It is clear to see that breathing on the metasurface, with both a hydrophobic and hydrophilic coating has a similar effect, which is also seen in the PVA metasurfaces. That is, the background light is increased dramatically, which results in the holographic image being more difficult to distinguish. We calculate the conversion efficiency of the a-Si metasurfaces to be around 0.1% for Meta A and Meta B, which is however, lower than that of the PVA metasurfaces (each being ~2% for 532 nm illumination). As can be seen from Supplementary Video 3, the PVA metahologram shows similar characteristics, however it appears to be hidden by the background at certain RH. As and the background noise of the holographic image is increased, and the desired image cannot be distinguished anymore. We hypothesize that there could be some interaction directly with the water molecules, along with the swelling, as well as the change in the refractive index of the PVA meta-atoms after absorbing water that all contribute to the PVA hologram producing an extremely low conversion efficiency (lower than 0.1% of the a-Si), making it disappear temporarily. This is under further investigation for future work, but we feel it goes a little too far outside of the scope of this work.

Figure R4. The influence of high RH conditions due to breathing on Pt-coated PVA metasurfaces. Comparison of holographic images produced from the master mold made of (i) a-Si with a hydrophobic coating, (ii) a-Si with a hydrophilic coating, and (iii) the hydrophilic Pt-coated PVA

To address this important comment from the reviewer, we have added the following to the revised manuscript and Supplementary Information:

In the manuscript:

Line 257: “When exposed to RH 80% using a breath, the hidden word 'PASSWORD' is decrypted temporarily due to the difference in swelling of Meta A and Meta B, while the holographic information is temporarily hidden due to the increased noise in the holographic image (Supplementary Note 17).”

In the Supplementary Information:

“Supplementary Note 17: Increased noise in the holographic image due to high RH

Figure S15. The influence of high RH conditions due to breathing on Pt-coated PVA metasurfaces. Comparison of holographic images produced from the master mold made of (i) a-Si with a hydrophobic coating, (ii) a-Si with a hydrophilic coating, and (iii) the hydrophilic Pt-coated PVA

“When the Pt-coated PVA metasurfaces are exposed to high RH conditions using a breath, the holographic image is temporarily hidden, while the background noise is dramatically increased. To analyze the reason for the disappearance of the holographic image, we investigate the holographic images produced by the amorphous silicon (a-Si) master mold. First, we treat it with a hydrophobic coating to help repel any water vapor, and then we treat with a hydrophilic coating, to emulate the hydrophilicity of the PVA metasurfaces. All three samples are exposed to the same high RH conditions using a breath and the holographic images are captured (Figure S15). It can be seen that during exposure to a breath, all three samples demonstrate a dramatically increased background noise to the holographic image. This can be attributed to water molecules forming on the metasurfaces which promote local scattering of the light. Although the background noise is increased, both holographic images from the a-Si samples are visible, whereas the Pt-coated PVA metasurface temporarily

displays no discernable image. We attribute this to the reduction in the refractive index of the PVA meta-atoms as well as increased scattering due to absorbing water molecules.”

Comment 5:

The authors claim that the hidden word is decrypted under the high RH, while the holographic image is hidden. After analyzing Supplementary Video 3, I think this looks misleading. From the video, it could be observed that the nanoprinting image and holographic image can appear/disappear simultaneously during breathing. The authors should carefully address the association between image channel switching and humidity tuning.

Response 5:

We thank the reviewer for their critical comment on ambiguous description of the reversible encryption system with Pt-coated PVA metasurfaces. The reviewer is correct that the reflected image appears simultaneously as the holographic image disappears. We see that the video that we included may be confusing, as we combined optical microscopy measurements with far field holographic images, by matching the response to the exhaled breath. In the revised Supplementary Videos 3 and 4, we have split the two channels into their own videos to avoid any confusion that could occur, since it is impossible for us to simultaneously measure both the color print and the holographic images in the current samples.

To give more clear description and avoid any unwanted confusion, we experimentally verified the critical RH where the color printed image can be decrypted. Along with our responses and extra experiments that we have undertaken to answer the other comments, especially Comment 4, we believe that the revised manuscript is much clearer and scientifically sound. A humidity sensor was integrated to quantify the relative humidity while simultaneously obtaining the images. The relative humidity was gradually increased from RH 58% to 82% by blowing N₂ through water vapor into the inlet (Figure R5a). The color print images are decrypted at RH 68% (Figure R5b). Considering that the measured RH of human breath is over 80% (as we have included in the revised Supplementary Information as Supplementary Note 14, and copied below (Figure R5) for the ease of the reviewer), the quantified threshold (of RH 68%) validates our encryption/decryption system using human breath. We have added the detailed explanation to both Supplementary Information and manuscript, and copied the additions below:

Figure R5. Measurement of the decryption threshold. (a) Measurement setup to quantify threshold of decryption. The chamber is uncovered in the figure for clarity, but it is covered in the experiment. (b) Optical images are obtained at i) RH 58%, ii) RH 68%, and iii) RH 82%. Scale bar: 50 μm .

In the Supplementary Information:

“Supplementary Note 14: Measured relative humidity of human breath

Figure S12. Measured RH of human breath. The RH of the room was measured to be 45%. Two different subjects breathed on the humidity sensor at a distance of 2.5 cm away for around 1.5 sec. The dots represent the maximum values of RH measured for each breath.

To confirm the value of RH that is achieved under the exhalation of a human breath, we measured the maximum RH using a commercial humidity meter (Testo, Testo 625). Two different researchers breathed on the meter at a distance of 2.5 cm away for around 1.5 sec, 10 times each, and the maximum RH reached was recorded (Figure S12). The RH of the room in which the experiment was conducted was measured to be 45%. The mean RH for subject 1 was measured to be 80.0%, while subject 2 was 83.9%. The measured values are comparable to the values quoted in the literature (85%)⁵.

In the manuscript:

Line 257: “When exposed to RH 80% using a breath, the hidden word 'PASSWORD' is decrypted temporarily due to the difference in swelling of Meta A and Meta B, while the holographic information is temporarily hidden due to the increased noise in the holographic image (Supplementary Note 17). Thanks to the Pt layer, on the contrary to the previous examples, the meta-atoms preserve their nanoscale geometry after exposure to RH 80%. This allows for the holographic image to be displayed again, and for the color print to be encrypted when the RH is reduced back to room level. The decryption is experimentally measured to occur when the RH is around 68% (Supplementary Note 18). The robustness of this reversibility is confirmed by exposing the PVA metasurface to exhaled breath up to 100 times (Fig. 5c).”

In the Supplementary Information:

“Supplementary Note 18: Measurement of the decryption threshold for the Pt coated PVA metasurfaces.

Figure S16. Measurement of the decryption threshold. (a) Measurement setup to quantify threshold of decryption. The chamber is uncovered in the figure for clarity, but it is covered in the experiment. (b) Optical images are obtained at i) RH 58%, ii) RH 68%, and iii) RH 82%. Scale bar: 50 μm .

The threshold RH to decrypt the hidden structural color image in the Pt-coated metasurfaces is measured using a humidity sensor implanted optical microscopy setup (Figure S16a). The sample had already been exposed to human breath 100 times to validate the repeatability of the optical encryption system. The RH is increased in the chamber by inserting N_2 gas to gradually increase the RH from 58% to 82%. The image is decrypted at RH 68%, and continues to be visible until around RH 82%, which correlates to the RH of human breath.”

Comment 6:

Regarding the geometry change of PVA nanostructure after being exposed to the high RH, the collapse of most structures (Figure 5c) would drastically influence the conversion efficiency. However, the holography seems to be unaffected. The authors should clarify this and add the discussion of the holographic efficiency.

Response 6:

We thank the reviewer for bringing up such critical points about our experimental results. As the reviewer points out, in Figure 5 it appears that most of structures had collapsed after the 1st breath onto the sample. However, somewhat unintuitively, the reconstructed holographic images appear to be only slightly influenced. To analyze the effect of the destruction of the meta-atoms on the quality of the reconstructed holographic images, we calculated the signal to noise ratio (SNR) of the holograms with different percentages of meta-atoms being destroyed (Figure R6). This gives us an idea of how the holographic image would look after some of the meta-atoms have collapsed, and no longer contribute the required phase at the desired spatial location. To do this, we assume when the structures have collapsed, then they have zero conversion efficiency. We select the meta-atoms at random across the metasurface. The SNR decreases from 53.1 to 31.14 db as the percentage of destroyed meta-atoms increases from 0 to 95%. Although the metasurfaces have only 5% of the meta-atoms contributing to the holographic image, the calculated SNR is 31.14 db, and the letters in the image are still fairly clearly identifiable. This is consistent with and analogous to various other examples of metasurfaces that have been demonstrated to exhibit multiple functionalities when interleaving meta-atoms for different phase maps, such as refs [R1-3]

Figure R6. Signal to noise ratio depending on percentage of destruction. SNR are calculated depending on different destruction percentage of (a) 0%, (b) 25%, (c) 50%, (d) 75%, (e) 90%, and (f) 95%. (i) Destructive parts are randomly assigned, and it is assumed

that the destroyed parts exhibit zero conversion efficiencies. (ii) Reconstructed images are calculated with computer generated phase map under destructions.

Furthermore, while working on the experiments to provide improved results for the revision of our manuscript, we amended the fabrication process of the deposition of the Pt coating on our PVA metasurface. Originally, we deposited the Pt coating at 15 mA for 60 s, but in the revised manuscript, we deposited the coating at 20 mA for 60 s. This has allowed us to improve the repeatability of our PVA metasurface, which we now experimentally demonstrate up to 100 repeated exposures to prove the robustness of our device. With the improved sample, we have completely new results for the Pt coated PVA metasurface, which has been updated in Figure 5. The new figure is copied below for the ease of the reviewer. The Pt coating now helps more of the meta-atoms to retain their geometries after exposure to high humidity conditions, as can be seen by comparing the SEM images of the as fabricated sample, with the sample after 100 exposures. This also helps to maintain the clarity of the holographic image compared to the previous results.

Figure 5:

“Figure 5. Reversible PVA metasurfaces for a two-channel security platform. (a) Schematic illustration of a Pt coated PVA meta-atom on top of a-Si:H. (b) Photographs of the holographic images under illumination of 532 nm coherent light, and reflected optical microscope images under white light illumination of the Pt coated PVA metasurfaces over 100 cycles of RH 80% exposure using human breath. Scale bars: 50 μm. (c) SEM images of the PVA metasurface (i) as fabricated, (ii) after 5 and (iii) 100 cycles of exposure to RH 80%. The nanostructures are maintained, allowing for the holographic information to be recovered. Scale bars: 1 μm.”

[R1] Nano Lett. 2016, **16**, 12, 7671–7676

[R2] Nano Lett. 2016, **16**, 8, 5235–5240

[R3] Optica 2017, **4**(11), 1368-1371

REVIEWERS' COMMENTS

Reviewer #1 (Remarks to the Author):

The authors have carefully revised the manuscript to address my concerns with the new data provided. Thus, I recommend this paper for publication in Nature Communications.

Reviewer #2 (Remarks to the Author):

The reviewer thanks the authors for the detailed response and comments. After carefully reviewing the response letter from the authors, I think the authors have addressed most of the mentioned minor concerns.

However, regarding the key issue – the novelty of the submitted work, it requires more critical clarification for considering to be published in Nature Comm.

The authors pointed out the key novelty and importance of this work because this is the first demonstration of nanoimprint lithography using hydrogels on large scale. However, I think it is important and necessary to make it clear why patterning hydrogels with nanoimprinting technique is difficult or challenging to realize in any aspect. I have not found any discussion on that. In my opinion, if patterning on hydrogel by imprint is not difficult but a trivial experiment in a conventional way which only has not been done at this point, then the novelty of the fabrication and combination of hydrogel and nanoimprinting does not sound quite strong enough for Nature Comm considering their previous work Ref [26] and other mentioned previous works.

Reviewer 1

The authors have carefully revised the manuscript to address my concerns with the new data provided. Thus, I recommend this paper for publication in Nature Communications.

Response to Reviewer 1**Response:**

We thank the reviewer for their time and effort in carefully reviewing our manuscript. We are extremely grateful for the influential suggestions of Reviewer 1 that helped us to dramatically improve the quality of our manuscript in the first revision, and for the support for publication in Nature Communications.

Reviewer 2

The reviewer thanks the authors for the detailed response and comments. After carefully reviewing the response letter from the authors, I think the authors have addressed most of the mentioned minor concerns.

Response to Reviewer 2**Response:**

First of all, we would like to give our appreciation to the reviewer for carefully evaluating our manuscript and providing us with such important and critical comments that have helped to further improve our manuscript through the revision process.

Comment:

However, regarding the key issue – the novelty of the submitted work, it requires more critical clarification for considering to be published in Nature Comm. The authors pointed out the key novelty and importance of this work because this is the first demonstration of nanoimprint lithography using hydrogels on large scale. However, I think it is important and necessary to make it clear why patterning hydrogels with nanoimprinting technique is difficult or challenging to realize in any aspect. I have not found any discussion on that. In my opinion, if patterning on hydrogel by imprint is not difficult but a trivial experiment in a conventional way which only has not been done at this point, then the novelty of the fabrication and combination of hydrogel and nanoimprinting does not sound quite strong enough for Nature Comm considering their previous work Ref [26] and other mentioned previous works.

Response:

We thank the reviewer for pointing out that we failed to successfully highlight the main novelties of our work in the revised manuscript. We will avoid repeating what we consider to be the main novelties of our work again in detail in this response letter, instead we will highlight each point through additions and revisions in the final manuscript, which we copy below:

To highlight the drawbacks of using hydrogels such as PVA in metasurfaces, and highlight how we can solve these problems with nanoimprinted PVA, we have added the following to the introduction:

Line 58

Using hydrogels such as PVA for optical metasurfaces provides another option for tunable, flat optical devices. However, to realize all of the benefits that metasurfaces offer, such as multiplexing various functionalities through phase and amplitude manipulation of the incident wavefront, the meta-atoms must first be tall enough to accumulate enough phase, and second must be nanoscale in size in order to operate at visible wavelengths.

Due to the fairly low refractive index of PVA, phase accumulation is quite low when fabricated at the subwavelength scales needed for metasurfaces. However, through our one-step nanoimprinting method, and exploration of the fabrication conditions that we provide in section 2.2, we are able to achieve meta-atoms with aspect ratios of ~ 10 , allowing us to use PVA for geometric phase metasurfaces for the first time. We have highlighted this fact by adding the following:

Line 148

This large aspect ratio opens up the potential to create meta-atoms that are tall enough to accumulate enough phase at the subwavelength scale, enabling the potential for local phase encoding that can be exploited for geometric phase metaholography.

In the conclusion, we have added the following to summarize and highlight the main novelties and contributions of our work:

Line 284

Furthermore, we utilized a one-step NIL technique to demonstrate all types of high-aspect ratio meta-atom structures using PVA, including nanogratings, nanopillars, and nanoholes, as well as demonstrating the freedom in substrate selection, including SiO_2 , a-Si:H, polymer, which could be useful for various applications where flexible, transparent, or strong substrates are required.

Line 290

We developed three different potential uses of such PVA metasurfaces, namely as single-use humidity sensitive smart labels and irreversible optical security, and by adding a thin layer of Pt, we demonstrated a method of creating a continuously reversible optical security device that is robust for at least 100 cycles, with almost no visible defects at the nanoscale as confirmed by SEM images.

Line 294

By choosing the appropriate substrate and coating, the reflective properties can be modulated, and the nanoscale geometry of the one-step imprinted meta-atoms can be selectively preserved or destroyed through exposure to high humidity conditions, providing a new alternative route for actively tunable metasurfaces that can manipulate the phase and amplitude of incident light at the subwavelength scale.